# RSAgent: Learning to Reason and Act via Multi-Turn Tool Invocations for Text-Guided Segmentation

**Xingqi He** [* 1]  **Yujie Zhang** [* 1 3]  **Shuyong Gao** [† 1]  **Wenjie Li** [3 4]  **Lingyi Hong** [1]  **Mingxi Chen** [1]  **Kaixun Jiang** [2] **Jiyuan Fu** [1]  **Wenqiang Zhang** [† 1 2]

## Abstract

Text-guided object segmentation requires both cross-modal reasoning and pixel grounding abilities. Most recent methods treat it as a single forward pass, where the model directly predicts pixel prompts to a segmentation model, which limits verification, refocusing and refinement when initial localization is wrong. To address this limitation, we propose **RSAgent**, an agentic Multimodal Large Language Model (MLLM) which interleaves **reasoning and action** for segmentation via multi-turn tool invocations. RSAgent queries a visual toolbox, observes feedback, and revises its pixel hypothesis using historical observations to re-localize targets and iteratively refine masks. We further build a data pipeline to synthesize multi-turn reasoning segmentation trajectories, and train RSAgent with a two-stage framework: cold-start supervised fine-tuning followed by agentic reinforcement learning with fine-grained, task-specific rewards. Extensive experiments show that RSAgent achieves a zero-shot performance of 66.5% gIoU on ReasonSeg test, improving over Seg-Zero-7B by 9%, and reaches 81.5% cIoU on RefCOCOg, demonstrating state-of-the-art performance on both in-domain and out-of-domain benchmarks.

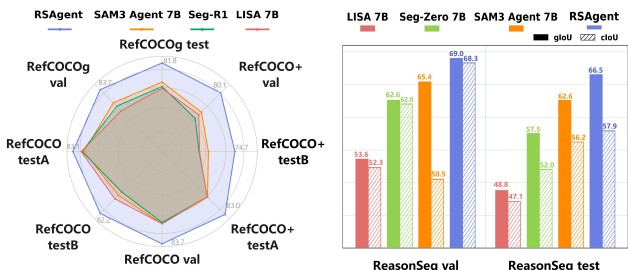

*Figure 1.* Performance on text-guided object segmentation benchmarks. RSAgent achieves state-of-the-art performance on both RES and ReasonSeg benchmarks.

## 1. Introduction

Text-guided segmentation takes an image and a natural language description of the task object as input and returns a fine-grained segmentation mask, covering both referring expression segmentation (RES) (Kazemzadeh et al., 2014; Yu et al., 2016) and reasoning segmentation (Lai et al., 2024), where the input text may be explicit expressions or implicit reasoning queries. Unlike closed-set semantic segmentation, which is restricted to a predefined label space, this setting must generalize to open-vocabulary concepts and therefore requires tight coupling between linguistic understanding and visual localization. This formulation introduces several key challenges: (i) grounding the task object to the correct region under large appearance variation, (ii) performing compositional or relational reasoning across modalities, and (iii) translating high-level linguistic cues into precise pixel boundaries. These requirements make text-guided segmentation particularly attractive for interactive perception and embodied applications (e.g., robotics).

Multimodal Large Language Models (MLLMs) have become a central component in the latest paradigms (Ren et al., 2024b; Bai et al., 2024; Rasheed et al., 2024; An et al., 2024; Nirkin et al., 2021; Xie et al., 2025; Wang et al., 2025a). Early attempts such as LISA (Lai et al., 2024), have explored the use of MLLMs to enhance reasoning segmentation capabilities via predefined semantic tokens (e.g., <SEG> token), bridging the gap between MLLMs and segmentation models by supervised fine-tuning (SFT).

---
[*]Equal contribution  [1]Shanghai Key Lab of Intelligent Information Processing, College of Computer Science and Artificial Intelligence, Fudan University [2]College of Intelligent Robotics and Advanced Manufacturing, Fudan University [3]Shanghai Innovation Institute [4]College of Health Science and Technology, Shanghai Jiao Tong University School of Medicine. Correspondence to: Shuyong Gao <shuyongg@andrew.cmu.edu>, Wenqiang Zhang <wqzhang@fudan.edu.cn>.

*Proceedings of the $43^{rd}$ International Conference on Machine Learning*, Seoul, South Korea. PMLR 306, 2026. Copyright 2026 by the author(s).

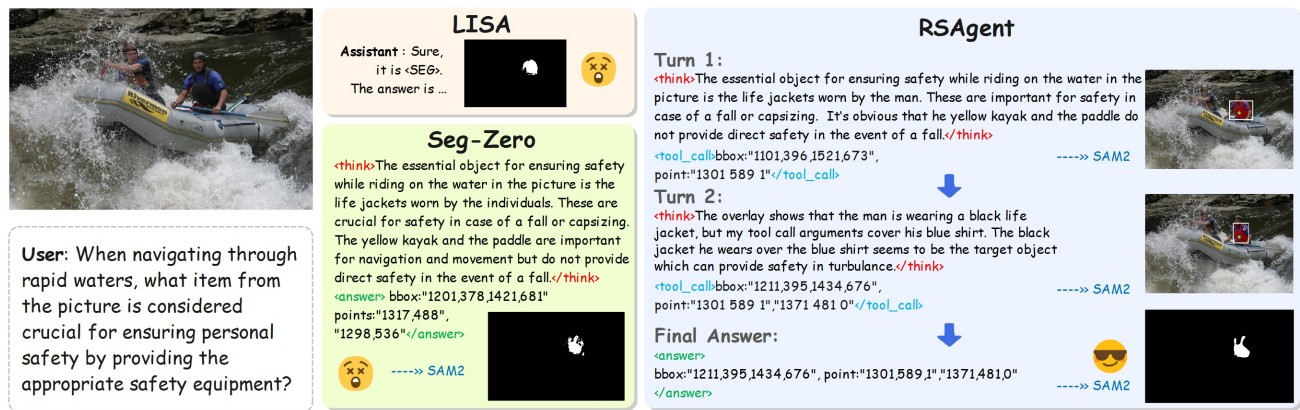

*Figure 2.* Comparing to LISA's direct segmentation (Lai et al., 2024) and Seg-Zero's single forward pass of thinking and segmentation (Liu et al., 2025a), RSAgent operates by iteratively proposing or updating pixel prompts and invoking visual tools to iteratively refine the mask.

Although SFT approaches can effectively inject MLLM reasoning into segmentation pipelines, they often face issues such as eroding general reasoning competence and weak robustness under distribution shifts. Reinforcement learning (RL) style approaches, meanwhile, optimize the model with reward signals for generating pixel prompts (e.g., boxes and points) and then feed them into Segment Anything Model (SAM) (Kirillov et al., 2023) to produce the mask. The reward design evolves from pixel prompt's accuracy (Liu et al., 2025a) to the predicted mask's IoU with ground truth (GT) mask (Huang et al., 2025b; You & Wu, 2025). However, these RL structures face two major limitations: (i) single prompt generation and directly obtaining the final reward not only completely forgoes SAM's inherent capability for iterative refinement, but also prevents the model from revisiting visual information it initially overlooked, particularly those seemingly irrelevant and ambiguous regions to initiate new segmentation attempts, and (ii) the reward function may over-emphasize coarse overlap or prompt-level accuracy, while providing limited supervision on reasoning process, which probably results in unexpected masks.

Therefore, to address these limitations, we present RSAgent, an agentic MLLM that performs interleaved cross-modal reasoning and action for segmentation. Specially, RSAgent reformulates text-guided segmentation as an interactive problem: as shown in Figure 2, given an image and a problem, the MLLM does not output mask tokens, nor relies on single-pass prompting, but instead produces coherent textual reasoning and decides tool actions to receive feedback for visual reflection. This iterative loop allows RSAgent to re-localize when early pixel hypotheses are wrong or refine the mask according to visual reflection. To enhance this agentic ability, we first introduce a data pipeline to generate multi-turn reasoning segmentation trajectories, constructing a high quality dataset including 5K complete reasoning trajectories and 2K image-problem pairs. Further we develop a two-stage training framework: a cold-start SFT phase

for model's proficiency in multi-turn tool invocations, followed by RL that optimizes long-horizon decisions with rewards that revisit while encouraging continuous mask improvement. As shown in Figure 1, RSAgent achieves state-of-the-art performance on both RES and ReasonSeg benchmarks, demonstrating that our approach is a stronger and more robust pathway for text-guided segmentation.

Our contributions can be listed as follows:

- We propose RSAgent with a framework of multi-turn tool invocations, which enables the MLLM to query a visual toolbox, obtain visual feedback and revise its pixel hypothesis using historical observations to re-localize targets and iteratively refine masks.

- We present a multi-turn reasoning trajectory generation pipeline, which helps to build the dataset for training. What's more, to enhance the agent's cross-modal reasoning ability, we introduce a two-stage training strategy: cold-start SFT and RL with fine-grained rewards.

- We conduct extensive in-domain and out-of-domain experiments to evaluate RSAgent. Results show that RSAgent achieves 81.5% average cIoU on RefCOCOg, 66.5% gIoU on ReasonSeg test, establishing new state-of-the-art performance for text-guided segmentation.

## 2. Related Work

**Reasoning in Large Language Models.** Recent LLMs have exhibited strong gains in deliberate reasoning: scaling inference time computation can reliably improve performance on challenging problems (OpenAI, 2024), and process-aware supervision or verification further strengthens step-by-step solution quality (Uesato et al., 2022; Lightman et al., 2023; Wang et al., 2024). Beyond purely text-only reasoning, reason-and-act paradigms such as ReAct (Yao et al., 2022) interleave natural language rationales with actions,

enabling models to revise intermediate hypotheses using external observations rather than relying solely on token histories. RL based post-training can further amplify these behaviors: DeepSeek-R1 (Guo et al., 2025) demonstrates that RL can incentivize long-form reasoning patterns including self-reflection and verification. However, most prior work primarily targets high-level correctness and coarse-grained multimodal understanding (Wang et al., 2025b; Hong et al., 2025), leaving dense, pixel-level reasoning tasks such as segmentation relatively underexplored. In contrast, RSAgent extends the "reason-and-act" paradigm to segmentation: it preserves coherent textual reasoning while interacting with a decoupled visual toolbox, enabling visual reflection for iterative re-localization and refinement.

**MLLMs for Text-Guided Segmentation.** Recent studies endow MLLMs with pixel-level segmentation capability from natural language prompts, typically by augmenting an MLLM with a mask generation module, e.g., a promptable segmentor such as SAM or a lightweight mask decoder. Early token-based approaches introduce special segmentation tokens (e.g., <SEG>) and an embedding-as-mask interface that maps language-model representations to dense masks (Lai et al., 2024; Bai et al., 2024; An et al., 2024; Wang et al., 2025a). Building on this line, MLLM-centric systems further move toward unified pixel grounding under conversational and multi-target settings, such as PixelLM (Ren et al., 2024b) and GLaMM (Rasheed et al., 2024). GSVA (Xia et al., 2024) extends referring segmentation to the generalized setting by supporting multiple targets and explicitly rejecting empty referents. Complementary to SFT style training, several recent methods emphasize RL for learning a decoupled policy that outputs pixel prompts to guide an external segmentor: Seg-Zero (Liu et al., 2025a) produces position prompts with RL, SAM-R1 (Huang et al., 2025b) leverages mask reward feedback for fine-grained alignment, and VisionReasoner (Liu et al., 2025b) unifies multiple perception tasks in an RL-based framework. Besides, POPEN (Zhu et al., 2025) explores preference-based optimization and ensembling to improve segmentation quality and reduce hallucinations.

## 3. Method

In this section, we present RSAgent, the data pipeline and the training strategies. We first formulate text-guided segmentation as an episodic decision making problem (Section 3.1). Then we provide an overview of RSAgent (Section 3.2). Next, we describe our data pipeline and cold-start SFT strategy (Section 3.3). Finally, we introduce the RL stage (Section 3.4), which further improves multi-round refinement and encourages efficient tool invocation.

### 3.1. Problem Formulation

Unlike prior approaches that predict a mask in a single forward pass of the model (Liu et al., 2025a; You & Wu, 2025), we cast text-guided segmentation as an episodic decision making problem in pixel space. As illustrated in Figure 3, the agent interacts with an external visual toolbox over multiple rounds, incrementally gathers visual evidence, refines candidate masks, and eventually commits to a final prediction. Formally, each example consists of an image $I$, a natural language problem $Q$ which could be either an explicit referring expression or a complex description, and, during training only, a GT mask $M_{\mathrm{gt}}$.

### 3.2. Overview of RSAgent

**Agent-tool interaction.** We formalize RSAgent as a vision-language policy $\pi_\theta$ operating in a finite-horizon Markov Decision Process (MDP). At step $t$, the agent receives an observation $O_t = (V_t, Q, C_t)$, where $V_t$ denotes the set of visual views available to the agent (including the original image and a history pool of overlays summarizing previous observations), and $C_t$ is the text context composed of the system prompt, the user prompts, and the accumulated assistant's feedback up to step $t$. For notational convenience, we write the reasoning trajectory up to step $T$ as:

$$P_{1:T} = \{(O_{t-1}, r_t, a_t, o_t)\}_{t=1}^{T}, \quad (1)$$

where $r_t$ denotes the model's intermediate reasoning tokens, $a_t$ denotes a set of parameterized tool calls and $o_t$ means the observation of this turn, which is concatenated to $O_t$. In the whole $P_{1:T}$, RSAgent alternates between textual reasoning, tool usage, and inspection of updated visual context, gradually refining its belief about the target object before committing to a final mask prediction. We provide a comprehensive notation example with description in Appendix A.

**Action space and tools.** Conditioned on $O_{t-1}$, the policy $\pi_\theta$ generates an interleaved sequence of reasoning and action tokens, which may contain (i) updated reasoning $r_t$ together with one or more structured tool invocations $a_t$, or (ii) a terminating answer of pixel prompts describing the final segmentation. The tools exposed to the agent form a compact but expressive action space: (i) view-manipulation operations that zoom in or rotate the image to adjust the field of view, and (ii) the major and frequently used segmentation operation that receives pixel prompts into candidate masks via a frozen SAM2 (Ravi et al., 2024). The episode terminates when the agent emits a dedicated answer or when a maximum interaction budget is reached (then the agent will be prompted to directly prompt the SAM2).

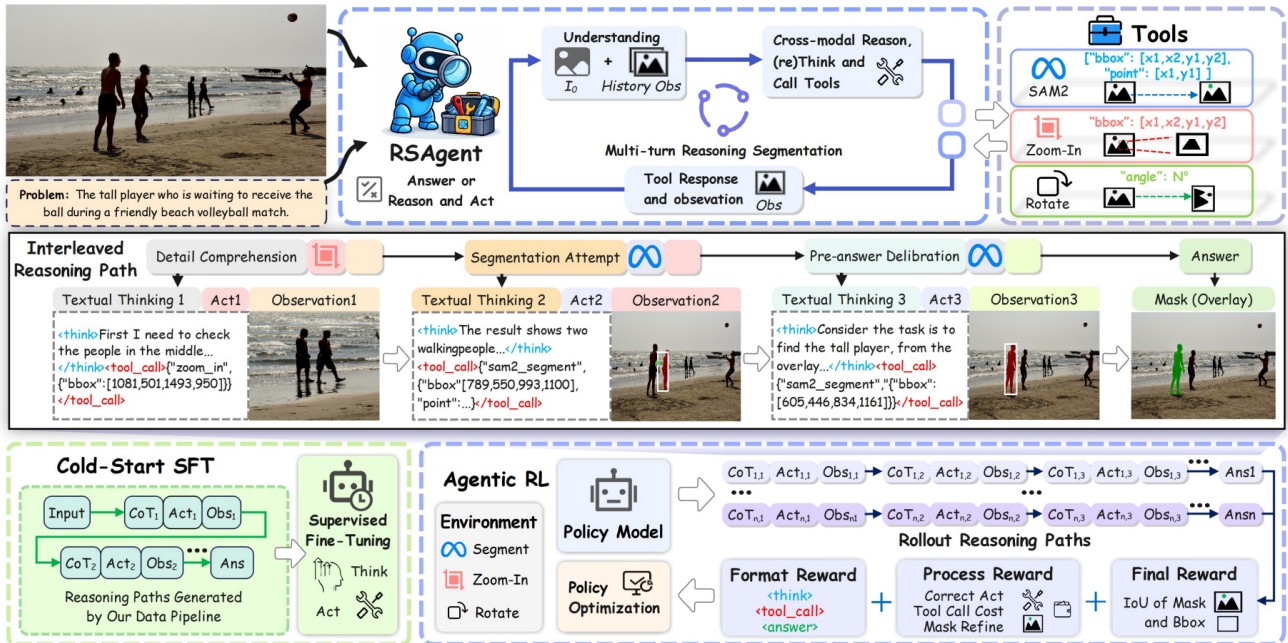

*Figure 3.* **Overview of RSAgent.** Given the original image and problem, the agent interacts with an external visual toolbox over multiple rounds, incrementally gathers visual evidence, refines candidate masks, and eventually commits to a final prediction. RSAgent first embraces cold-start SFT to get accustomed to reasoning and multi-turn tool invoking operations via the cold-start data generated by our data pipeline, then gets optimized by RL with fine-grained rewards.

## 3.3. Data Pipeline and Cold-Start Supervised Fine-tuning

**Problem-centric data collection.** To synthesize the multi-turn reasoning trajectories, constructing the textual problems for images is the primary task. We first collect image–mask pairs $(I, M_{gt})$ and associated annotations from SA-1B dataset (Kirillov et al., 2023). Then we construct a natural language problem $Q$ describing the target object. Concretely, we employ strong proprietary vision-language models (practically, we used Gemini2.5-Pro (Google, 2025) and OpenAI-o3 (OpenAI, 2025)) to generate problems in the style of reasoning segmentation according to $(I, M_{gt})$. The details are provided in Appendix B.

**Multi-turn tool interaction for trajectory synthesis.** After problem generation, we synthesize full multi-round trajectories with the seed pair $(I, Q)$ by letting a model with the ability of pixel understanding interact with the same tool environment as RSAgent and generate multi-turn interleaved reasoning trajectories. Concretely, we place Qwen2.5-VL-72B-Instruct (Bai et al., 2025) in our visual tool environment, which exposes the view manipulation and segmentation tools described in Section 3.2. The resulting interaction yields a full pixel-space reasoning trajectory with multi-turn tool calls and corresponding visual states. The overall pipeline is illustrated in Figure 4.

**Trajectory filtering.** To ensure that the synthetic trajectories provide reliable supervision, we select the cold-start data with two hard principles: (i) the IoU of the final predicted mask and $M_{gt}$ can't be lower than 0.9, and (ii) the number of reasoning turns should not surpass 6. The former principle not only validates the accuracy of the problem, but also demonstrates the multi-turn interleaved reasoning process's correctness. For the latter, although some trajectory's final answer is correct, the actual reasoning process may be unreliable due to overly long context, compounding errors or hallucinated rationales.

In addition to the high quality trajectories selected under the above criteria, we also filter a supplemental set of trajectories that exhibit a modest number of tool invocations and a low final mask IoU, for which a correct mask is produced by segmentation tool at some intermediate reasoning step. After manually verifying the correctness of the corresponding problem $Q$, we revise the terminal supervision to align with this correct intermediate mask and discard the most misleading portions of the erroneous reasoning trace, and then add these curated trajectories to our dataset.

**Cold-start SFT.** Through our data pipeline, we obtain a dataset $D_{sft}$ of approximately 5K examples, each containing both coherent reasoning trajectories and corrective refinement traces for cold-start SFT. The training objective is to minimize the average negative log-likelihood over all reasoning and action tokens in the cold-start trajectories.

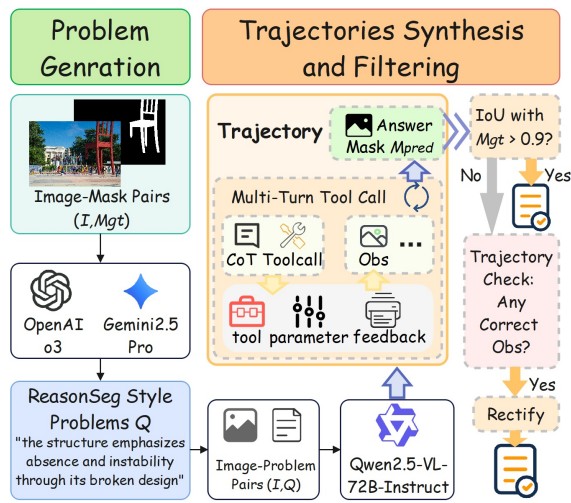

*Figure 4.* The multi-turn reasoning segmentation data pipeline, including problem generation, trajectory synthesis and data filtering.

Concretely, given a trajectory $P$ with token sequence $y_{1:|P|}$, we define a token mask $m_t \in \{0, 1\}$ to select reasoning and tool-call tokens, and optimize:

$$\mathcal{L}_{\text{SFT}}(\theta) = \mathbb{E}_{P \sim \mathcal{D}_{\text{sft}}} \left[ -\frac{1}{\sum_{t=1}^{|P|} m_t} \sum_{t=1}^{|P|} m_t \log \pi_\theta(y_t \mid y_{<t}) \right]. \quad (2)$$

This objective teaches the model to produce effective trajectories with robust step-by-step reasoning patterns, thereby endowing RSAgent with stronger pixel-level reasoning and self-correction capabilities and providing a solid foundation for subsequent RL training.

### 3.4. Agentic Tool Reinforcement Learning

We further optimize RSAgent with RL under a carefully designed reward scheme, enabling the agent to adaptively discover effective tool-use strategies for courageous attempt and iterative mask refinement.

#### 3.4.1. RL DATASET

The essence of RL training is to let the policy learn how to select actions that yield higher reward through interaction with the environment. Consequently, the RL dataset does not need to contain pre-generated multi-round trajectories: it is sufficient to provide the policy with the image $I$ and its associated problem $Q$ as input, together with the GT mask $M_{\text{gt}}$ used solely for reward computation.

We construct approximately 2K RL examples using the same data curation procedure as in Section 3.3, and additionally sample 8K instances from the RefCOCOg training split to form the overall RL dataset $D_{\text{rl}}$.

#### 3.4.2. REWARD DESIGN

Unlike prior approaches (Liu et al., 2025a; Huang et al., 2025b) that rely solely on outcome-based rewards for models that predict a segmentation mask in a single forward pass, we introduce a fine-grained reward design that jointly accounts for both the final result and the intermediate decision process, enabling the policy to acquire more efficient and effective decision strategies. Specifically, our reward function consists of the following components:

- **Final-answer reward** $R_{\text{final}}$: encourages high quality final masks via both mask IoU and bounding box IoU with the GT mask and bounding box.

- **Format reward** $R_{\text{format}}$: rewards syntactically valid, schema-compliant `<answer>` and `<tool_call>` blocks, penalizes unparsable outputs and invalid tool invocations.

- **Process reward** $R_{\text{process}}$: step-wise shaping based on IoU improvement, including the best-so-far IoU in the reasoning process, tool-dependent call costs, and point-level sparsity or novelty for segmentation prompts.

The unified reward function is formulated as:

$$R_{\text{total}} = \alpha \cdot R_{\text{final}} + \beta \cdot R_{\text{process}} + \gamma \cdot R_{\text{format}}, \quad (3)$$

where $\alpha, \beta, \gamma$ controls the relative strength of dense shaping versus outcome quality. The details of each reward are provided in Appendix C.

#### 3.4.3. TRAINING OBJECTIVE

Based on the rollout formulation in Section 3.2 and rewards defined above, we optimize the policy with Group Relative Policy Optimization (GRPO) (Shao et al., 2024) (without an explicit KL penalty term (Hu et al., 2025)) on dataset $\mathcal{D}_{\text{rl}}$. Let $y_{i,1:T_i}$ denote the concatenation of all non-observation tokens (reasoning tokens and tool-call tokens) generated in rollout $P_i$. The GRPO objective is:

$$\mathcal{L}_{\text{RL}}(\theta) = \mathbb{E}_{\substack{(I,Q,M_{\text{gt}}) \sim \mathcal{D}_{\text{rl}} \\ \{P_i\}_{i=1}^{G} \sim \pi_{\theta_{\text{old}}}(\cdot|I,Q)}} \left( -\frac{1}{G} \sum_{i=1}^{G} \frac{1}{T_i} \sum_{n=1}^{T_i} \right. \quad (4)$$
$$\left. \min\big(\rho_{i,n} A_i, \ \text{clip}(\rho_{i,n}, 1-\epsilon, 1+\epsilon) A_i\big) \right),$$

where $\rho_{i,n}$ denotes the importance ratio:

$$\rho_{i,n} = \frac{\pi_\theta(y_{i,n} \mid y_{i,<n}, I, Q)}{\pi_{\theta_{\text{old}}}(y_{i,n} \mid y_{i,<n}, I, Q)}. \quad (5)$$

Here, $G$ is the number of sampled rollout paths; $T_i$ is the number of non-observation tokens in $P_i$; and $S_i$ is the

*Table 1.* Comparison with previous state-of-the-art methods on RES benchmarks. We report cIoU (%) on RefCOCO, Ref-COCO+ (Kazemzadeh et al., 2014) and RefCOCOg (Yu et al., 2016). The overall best performances are shown in **bold**, while the second best performances are shown underlined.

| Method | Model | | RES | RefCOCO | | | RefCOCO+ | | | RefCOCOg | |
| | Version | Params | **Train** | val | testA | testB | val | testA | testB | val | test |
|---|---|---|---|---|---|---|---|---|---|---|---|
| *Discriminative vision–language segmentors without LLM-based controllers.* | | | | | | | | | | | |
| LAVT | – | – | Y | 72.7 | 75.8 | 68.8 | 62.1 | 68.4 | 55.1 | 61.2 | 62.1 |
| ReLA | – | – | Y | 73.8 | 76.5 | 70.2 | 66.0 | 71.0 | 57.7 | 65.0 | 66.0 |
| DETRIS | DETRIS-L | – | Y | 81.0 | 81.9 | 79.0 | 75.2 | 78.6 | 70.2 | 74.6 | 75.3 |
| UniLSeg | UniLSeg-100 | – | Y | 81.7 | 83.2 | 79.9 | 73.2 | 78.3 | 68.2 | 79.3 | 80.5 |
| EVF-SAM | Extra Data | – | Y | 82.4 | 84.2 | 80.2 | 76.5 | 80.0 | 71.9 | 78.2 | 78.3 |
| *Large (multimodal) models for single-pass segmentation without explicit reasoning traces.* | | | | | | | | | | | |
| LISA | LLaVA1.5 | 7B | Y | 74.9 | 79.1 | 72.3 | 65.1 | 70.8 | 58.1 | 67.9 | 70.6 |
| GSVA | LLaVA1.5 | 13B | Y | 79.2 | 81.7 | 77.1 | 70.3 | 73.8 | 63.6 | 75.7 | 77.0 |
| GLaMM | Vicuna | 7B | Y | 79.5 | 83.2 | 76.9 | 72.6 | 78.7 | 64.6 | 74.2 | 74.9 |
| SAM4MLLM | LLaVA1.6 | 8B | Y | 79.8 | 82.7 | 74.7 | 74.6 | 80.0 | 67.2 | 75.5 | 76.4 |
| PixelLM | LLaVA2 | 7B | Y | 73.0 | 76.5 | 68.2 | 66.3 | 71.7 | 58.3 | 69.3 | 70.5 |
| RICE | Qwen2.5 | 7B | Y | 83.5 | **85.3** | 81.7 | 79.4 | 82.8 | **75.4** | 79.8 | 80.4 |
| *Explicit chain-of-thought (CoT) or reasoning-chain guided segmentation.* | | | | | | | | | | | |
| Seg-Zero | Qwen2.5-VL | 7B | Y | – | 80.3 | – | – | 76.2 | – | 72.6 | – |
| Seg-R1 | Qwen2.5-VL | 7B | N | 74.3 | 78.7 | 67.6 | 62.6 | 70.9 | 57.9 | 71.0 | 71.4 |
| *RSAgent-single* | Qwen2.5-VL | 7B | Y | 80.6 | 81.1 | 78.9 | 77.3 | 81.1 | 70.4 | 73.7 | 76.0 |
| *Multi-round tool-calling agents with iterative refinement.* | | | | | | | | | | | |
| SAM3 Agent | Qwen2.5 | 7B | N | 53.4 | 58.4 | 48.0 | 46.3 | 52.2 | 40.8 | 54.5 | 55.1 |
| SAM3 Agent | Gemini2.5-Pro | – | N | 74.9 | 77.8 | 69.9 | 66.9 | 71.1 | 62.4 | 73.3 | 73.6 |
| *RSAgent (ours)* | Qwen2.5-VL | 7B | Y | **83.7** | 83.1 | **82.2** | **80.1** | **83.0** | 74.7 | **81.3** | **81.8** |

trajectory-level return computed from our reward. We compute a group-relative advantage $A_i$ by normalizing returns within each rollout group:

$$A_i = \frac{S_i - \mu_S}{\sigma_S}, \qquad \mu_S = \frac{1}{G} \sum_{j=1}^{G} S_j,$$

$$\sigma_S = \sqrt{\frac{1}{G} \sum_{j=1}^{G} (S_j - \mu_S)^2 + \delta}, \qquad (6)$$

where $\delta$ is a small constant for numerical stability. Through carefully designed RL training, the agent becomes capable of interpreting target descriptions and performing pixel-level reasoning via iterative tool invocations, thereby accurately segmenting the object in the image that best matches the textual problem.

## 4. Experiment

### 4.1. Experimental Settings

**Implementation Details.** We implement our RSAgent based on Qwen2.5-VL-7B-Instruct (Bai et al., 2025). SAM2-large (Ravi et al., 2024) is used as the segmentation tool. In cold-start SFT stage, we optimize the model with a learning rate of $2 \times 10^{-5}$ for 2 epochs and the batch

size is 128. Then we implement the RL optimization on VERL (Sheng et al., 2025) framework with batch size of 16 and sampling number of 4. The max prompt length is 32K and learning rate is $1 \times 10^{-6}$. Moreover, the max number of tool call turns is 8. Whether in RL training or evaluation, answer will be given if the number of turns exceeds it.

**Benchmarks and Datasets.** We mainly conduct experiments on two benchmarks: RES benchmarks (RefCOCO series, which includes RefCOCO, RefCOCO+ and RefCOCOg (Kazemzadeh et al., 2014; Yu et al., 2016)) and ReasonSeg (Lai et al., 2024) benchmarks. To ensure fair evaluation and comparison with previous work, we only randomly sampled 8K data from RefCOCOg train split as part of $D_{rl}$ and keep ReasonSeg dataset out-of-domain. The 5K dataset $D_{sft}$ and the rest of 2K data in $D_{rl}$ are all generated with random samples from SA-1B by the data pipeline introduced in Section 3.3.

**Evaluation Metrics.** Following previous work on text-guided segmentation, we adopt two commonly used evaluation metrics: generalized IoU (gIoU) and cumulative IoU (cIoU). Specifically, gIoU is computed as the average of per-image IoU, while cIoU is defined as the IoU of the cumulative predicted and GT masks across the entire dataset.

*Table 2.* Comparison with state-of-the-art methods on the ReasonSeg benchmark (Lai et al., 2024). We report generalized IoU (gIoU) and cumulative IoU (cIoU) in %. The overall best performances are shown in **bold**, while the second best performances are shown underlined.

| Method | Model | | Training | | Val | | Test | | Test (Short) | | Test (Long) | |
|---|---|---|---|---|---|---|---|---|---|---|---|---|
| | Version | Params | RES | ReasonSeg | gIoU | cIoU | gIoU | cIoU | gIoU | cIoU | gIoU | cIoU |
| *Discriminative vision–language segmentors without LLM-based controllers.* | | | | | | | | | | | | |
| SEEM | – | – | N | N | 25.5 | 21.2 | 24.3 | 18.7 | 20.1 | 11.5 | 25.6 | 20.8 |
| Grounded-SAM | – | – | N | N | 26.0 | 14.5 | 21.3 | 16.4 | 17.8 | 10.8 | 22.4 | 18.6 |
| OVSeg | – | – | N | N | 28.5 | 18.6 | 26.1 | 20.8 | 18.0 | 15.5 | 28.7 | 22.5 |
| *Large (multimodal) models for single-pass segmentation without explicit reasoning traces.* | | | | | | | | | | | | |
| GLaMM | Vicuna | 7B | Y | N | 47.4 | 47.2 | – | – | – | – | – | – |
| SAM4MLLM | LLaVA1.6 | 8B | Y | N | 58.4 | 60.4 | – | – | – | – | – | – |
| X-SAM | Phi-3 | 3.8B | Y | Y | 56.6 | 32.9 | 57.8 | 41.0 | 47.7 | 48.1 | 56.0 | 40.8 |
| HyperSeg | Phi-2 | 3B | Y | Y | 59.2 | 56.7 | – | – | – | – | – | – |
| LISA | LLaVA1.5 | 7B | Y | N | 53.6 | 52.3 | 48.8 | 47.1 | 48.3 | 48.8 | 49.2 | 48.9 |
| LISA | LLaVA1.5 | 7B | Y | Y | 61.3 | 62.9 | 51.7 | 51.1 | 44.3 | 42.0 | 57.9 | 59.7 |
| RICE | Qwen2.5 | 7B | Y | N | 58.8 | 55.9 | 57.3 | 54.6 | – | – | – | – |
| *Explicit chain-of-thought (CoT) or reasoning-chain guided segmentation.* | | | | | | | | | | | | |
| RSVP | LLaVA1.6 | 7B | N | N | 59.2 | 56.7 | 56.9 | 50.7 | 47.9 | 42.0 | 58.4 | 53.0 |
| RSVP | GPT-4o | – | N | N | 64.7 | 63.1 | 60.3 | **60.0** | 55.4 | 50.4 | 61.9 | **62.5** |
| Seg-Zero | Qwen2.5-VL | 7B | Y | N | 62.6 | 62.0 | 57.5 | 52.0 | – | – | – | – |
| Seg-R1 | Qwen2.5-VL | 7B | N | N | 58.6 | 41.2 | 56.7 | 53.7 | – | – | – | – |
| SAM-R1 | Qwen2.5-VL | 7B | Y | N | 64.0 | 55.8 | 60.2 | 54.3 | – | – | – | – |
| *RSAgent-single* | Qwen2.5-VL | 7B | Y | N | 58.2 | 53.1 | 56.1 | 52.6 | 54.9 | 47.7 | 62.1 | 56.3 |
| *Multi-round tool-calling agents with iterative refinement.* | | | | | | | | | | | | |
| SAM3-Agent | Qwen2.5-VL | 7B | N | N | 65.4 | 50.5 | 62.6 | 56.2 | 59.1 | 41.8 | 63.7 | 57.8 |
| *RSAgent (ours)* | Qwen2.5-VL | 7B | Y | N | **69.0** | **68.3** | **66.5** | 57.9 | **60.3** | **54.4** | **68.4** | 60.1 |

**Baselines.** We compare RSAgent with four families of representative methods: (i) discriminative vision–language segmentors without LLM-based controllers, (ii) Large (multimodal) models for single-pass segmentation without explicit reasoning traces, (iii) Explicit chain-of-thought (CoT) or reasoning-chain guided segmentation, and (iv) multi-round tool-calling agents with iterative refinement. We restrict RSAgent to output mask in single forward pass as *RSAgent-single* as additional of our baselines. The detailed information of our baselines is introduced in Appendix D.

### 4.2. Main Results

**Results on Referring Segmentation Benchmarks.** As shown in Table 1, RSAgent achieves competitive performance on RefCOCO and RefCOCO+, and particularly state-of-the-art results on RefCOCOg. Despite being trained on only a 8K subset of RefCOCOg, RSAgent surpasses several MLLM-based baselines built on comparable or larger backbones like RICE (Xie et al., 2025), outperforms the CoT guided single-pass segmentation models such as Seg-Zero (Liu et al., 2025a) and substantially surpasses SAM3 Agent (Carion et al., 2025) which only utilizes prompt engineering without training. What's more, RSAgent-single achieves competitive performance and even outperforms the single-pass segmentation models, demonstrating our

method successfully enhanced the model's reasoning ability. As previous work tell the GT annotations in RefCOCO (+/g) are not precise enough, which suggests that our RSAgent model should, in principle, achieve better performance than values in the table. We attribute these gains to two factors: (i) the multi-round tool invocation, which allows RSAgent to zoom in, segment and iteratively refine segmentation proposals conditioned on the action outcome, and (ii) the cold-start SFT which enable the model to reason-and-act and RL objective that explicitly rewards mask improvement and penalizes unnecessary tool calls.

**Results on Reasoning Segmentation Benchmarks.** As shown in Table 2, our RSAgent achieves state-of-the-art performance in ReasonSeg test and val in gIoU and competitive results in other splits. Quantitatively, RSAgent considerably outperforms earlier SAM2-based single-pass baselines such as (Liu et al., 2025a; Huang et al., 2025b) by at least 5.0% gIoU on val split, 6.3% cIoU on val split, 6.3% gIoU on test split and 3.6% cIoU on test split, as it can handle long and compositional descriptions involving pixel relations or fine-grained attributes more robustly by focusing on ambiguous regions before committing to a final mask. Similarly, RSAgent-single's performance also outperforms the previous CoT guided single-pass segmentation models. Although SAM3 Agent drives SAM3 model to achieve strong perfor-

*Table 3.* Ablation on cold-start SFT and RL. Metrics are cIoU (%).

| Setting | ReasonSeg test | RefCOCOg testA |
|---|---|---|
| single-pass baseline (1-pass) | 43.9 | 64.9 |
| tool-agent (no training) | 30.1 | 61.7 |
| + cold-start SFT only | 55.4 | 73.9 |
| + RL only | 54.3 | 77.3 |
| + cold-start SFT + RL (full) | **57.9** | **81.8** |

mance on ReasonSeg through training-free multi-round tool invocations, RSAgent, equipped with efficient SFT and RL, can likewise operate in a multi-round tool-calling regime to control a SAM2-based toolbox and attain comparable, or even superior, evaluation results despite relying on the comparatively weaker SAM2 backbone, which reflects more holistic segmentation quality across objects and scenes.

**Qualitative Analysis.** According to the quantitative analysis and the qualitative exhibition of Figure 2, existing segmentation approaches that rely on implicit tokens distort the native textual output space of MLLMs, thereby compromising their language capability and weakening semantic generalization, while single-pass of segmentation methods lack progressive mask refinement ability. In contrast, RSAgent not only generates correct and coherent textual reasoning, but also performs visually grounded reflection through multi-turn tool invocations. This interleaved reasoning ability stems from our design to decouple reasoning from a segmentation-centric tool environment.

### 4.3. Ablations

We conduct several ablation studies to verify the effectiveness of our design. The default settings are as follows: (i) we keep the base model, dataset and the training parameters the same as the RSAgent and (ii) except for the segmentor ablation study, the segmentation tool is SAM2-hiera-large.

**Analysis of cold-start SFT and RL.** We first verify whether cold-start SFT and RL are indispensable components of our training pipeline. As shown in Table 3, directly deploying an off-the-shelf Qwen2.5-VL-7B-Instruct model without any additional training in our framework yields poor performance compared with single forward pass baselines. We attribute this to a paradigm mismatch: the base model is primarily pretrained for QA-style generation and is not explicitly exposed to tool-use interactions, leading to significant misalignment when transferred to a novel tool invocation setting. In contrast, after cold-start SFT (or even RL alone), the model learns to invoke external tools to support interleaved reasoning segmentation.

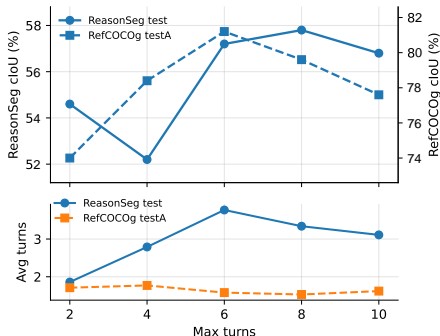

*Figure 5.* Effect of the maximum number of tool-invocation turns during training. Metrics are cIoU (%).

*Table 4.* Ablation on reward components. Metrics are cIoU (%).

| Reward Variant | ReasonSeg test | RefCOCOg testA |
|---|---|---|
| Full ($R_{\text{final}} + R_{\text{process}} + R_{\text{format}}$) | **57.9** | **81.8** |
| w/o $R_{\text{final}}$ | 48.3 | 68.1 |
| w/o $R_{\text{process}}$ | 55.9 | 77.6 |
| w/o $R_{\text{format}}$ | 54.7 | 80.3 |

**Analysis of multi-turn tool invocation.** Although multi-turn of decision in segmentation helps the model to rethink and provide accurate masks, overlong or unnecessary context may results in unexpected masks. As shown in Figure 5, the model's performance gradually improves as the max number of turns to call tools during training (max-turns) gets higher until 8 turns. What's more, during inference the average turns of successful predictions (when IoU > 0.9) stops until max-turn reaches 6. For RefCOCOg benchmark which is less challenging compared with ReasonSeg, the tendency comes out the same, which indicates that appropriate max-turns during training matters and RSAgent acquires an efficient adaptive stopping rule.

**Analysis of reward design.** We explore the impact of different reward functions during training. We study the model trained without $R_{\text{final}}$, $R_{\text{process}}$ and $R_{\text{format}}$. Table 4 shows that removing either component leads to consistent performance drops, demonstrating every reward's unique role in multi-turn reasoning segmentation ability construction.

*Table 5.* Ablation on segmentation tools. Metrics are cIoU (%).

| Segmentation Model | ReasonSeg test | RefCOCOg testA |
|---|---|---|
| SAM2-hiera-large | **57.9** | **81.8** |
| SAM2-hiera-base | 56.7 | 80.9 |
| SAM2-hiera-tiny | 53.1 | 77.2 |
| SAM-H | 53.3 | 74.6 |
| HQ-SAM-H | 52.9 | 75.1 |

**Analysis of segmentation tool.** We further evaluate RSAgent's reasoning segment ability with different versions of SAM and HQ-SAM (Ke et al., 2023). Table 5 shows that RSAgent consistently benefits from iterative interaction across tools, and can effectively exploit stronger segmentors while remaining robust when the tool is lightweight.

## 5. Conclusion

We propose RSAgent, an agentic MLLM for text-guided segmentation that predicts masks via multi-turn reason-and-act with an external toolbox. Given an original image and textual problem, RSAgent operates by generating pixel prompts and invoking visual tools to focus on the correct region and iteratively refine the final mask. To enable the MLLM with the ability, we first develop a data pipeline and generate a dataset of 7K samples for training. Then we develop a two stage framework specialized for the multi-turn tool invocations. Qualitative results and quantitative evaluations demonstrate that RSAgent achieves state-of-the-art performance in both in-domain and out-of-domain benchmarks, revealing its interleaved reasoning segmentation ability to trial and refine via multi-turn tool invocations.

## Software and Data

The project page is available at: `https://github.com/Nicola777-ai/RSAgent`.

At the time of publication, we provide the main implementation details, model and tool configurations, training recipes, reward definitions, hyperparameters, and evaluation protocols in the paper and appendix to facilitate reproducibility. All evaluation datasets used in this work are public benchmarks, including RefCOCO, RefCOCO+, RefCOCOg, and ReasonSeg. The repository will be used to maintain project-related materials and updates.

## Acknowledgements

This work was supported by National Natural Science Foundation of China (No.62576109), Scientific and Technological innovation action plan of Shanghai Science and Technology Committee (No.25511104402). We are grateful to all co-authors for their contributions, discussions, and collaboration throughout this project. We also thank the anonymous reviewers and the area chair for their constructive feedback and thoughtful suggestions.

## Impact Statement

Our work aims to strengthen cross-modal reasoning to improve text-guided segmentation performance. By enabling multi-turn perception and structured interaction, RSAgent may reduce manual trial-and-error in text-guided segmentation and benefit applications such as data annotation, robotic perception, and scientific or medical image analysis. As with other segmentation systems, outputs can be imperfect and may vary across domains or imaging conditions, so RSAgent should be used with appropriate validation and human oversight in high-stakes settings. The same capability could also be used to extract or highlight objects in personal images; we encourage responsible deployment that respects privacy, consent, and dataset licenses. We also recommend reporting training, evaluation details and compute to support reproducibility and more efficient reuse.

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

# A. Supplementary of RSAgent's Overview

**Pixel-space memory and trajectory.** When a segmentation tool is invoked, the environment selects one or more candidate masks and renders them back onto the original image as lightweight overlays (we choose the $384 \times 384$ as history pool's image size). These overlays are compressed into a fixed-size history pool and aggregated into $V_{t+1}$, so that subsequent decisions can attend jointly to the raw image and prior hypotheses.

**Termination and learning signal.** The episode terminates when the agent emits a dedicated answer segment specifying which candidate masks constitute the final output (optionally accompanied by a textual justification), or when a maximum interaction budget is reached. At training time, the environment evaluates the final prediction $\hat{M}$ against $M_{\mathrm{gt}}$ and provides a segmentation-aware return that combines the terminal IoU-based score with dense, step-wise shaping terms reflecting mask improvement, sensible tool usage, and trajectory efficiency. This unified agent–tool formulation underlies both the cold-start supervised fine-tuning in Section 3.3 and the reinforcement learning stage in Section 3.4.

*Table 6.* Notation used in RSAgent. Here, $r_t$ denotes the intermediate reasoning text generated by the model, rather than the scalar reward used in reinforcement learning.

| Symbol | Meaning in RSAgent | Concrete example in one episode |
|---|---|---|
| $I$ | Input image | An image containing a person holding a cup. |
| $Q$ | Natural-language segmentation query | "Segment the cup held by the person on the left." |
| $O_t$ | Observation at step $t$, $O_t = (V_t, Q, C_t)$ | At $t = 2$, the agent sees the original image, a zoomed crop from the previous step, the query $Q$, and the accumulated text/tool history so far. |
| $V_t$ | Visual views available at step $t$ | The original image, an overlay showing the previous SAM2 mask, and a zoomed-in crop around the candidate region. |
| $C_t$ | Text context accumulated up to step $t$ | The system prompt, user query, previous reasoning text, tool calls, and tool feedback. |
| $r_t$ | Intermediate reasoning text generated by the model at step $t$; this is not the scalar RL reward | "The cup is small and partially occluded; I should zoom into the hand region before segmenting." |
| $a_t$ | Parameterized tool action(s) at step $t$ | `zoom_in(bbox=[x1,y1,x2,y2])` or `sam2_segment(points=[...], labels=[...])`. |
| $o_t$ | Tool output / new visual feedback returned at step $t$ | A zoomed image patch, a rotated image, or a candidate mask produced by SAM2. |
| $P_{1:T}$ | Full multi-turn trajectory up to termination | A sequence of observations, reasoning text, tool calls, and returned tool outputs across all turns. |

**Example multi-turn trajectory.** As shown in Table 6, a typical RSAgent trajectory follows an observe–reason–act–feedback loop. At the first step, the agent receives $O_1 = (V_1, Q, C_1)$, where $V_1$ contains the original image and $C_1$ contains the initial system and user prompts. The model generates intermediate reasoning text $r_1$, (e.g., The queried cup is small and attached to the left person; I should first zoom into the left-hand area), and then emits a tool action $a_1 = $ `zoom_in(bbox)`.

At the second step, the toolbox returns $o_1$, namely a zoomed crop, which is incorporated into the next observation $O_2$. Conditioned on $O_2$, the model generates new reasoning text $r_2$ and emits a segmentation action such as $a_2 = $ `sam2_segment(points, labels)`.

At the third step, the toolbox returns $o_2$, e.g., a candidate mask overlay, which becomes part of $O_3$. The model can either refine the mask through additional tool calls or terminate when the result is satisfactory. Therefore, the full trajectory $P_{1:T}$ records the multi-turn process of observing visual/textual context, reasoning about the target, invoking external visual tools, receiving feedback, and finally outputting the segmentation mask.

# B. Details of Our Data Collection

To obtain supervision that is closely aligned with our multi-round RL environment, we construct a synthetic dataset of multi-step segmentation trajectories on top of a curated subset of SA-1B. The overall pipeline is illustrated in Figure 4.

## B.1. Problem-centric Data Collection

We build our cold-start data on top of the SA-1B dataset (Kirillov et al., 2023), a large-scale corpus of about 11M diverse images with 1.1B generated segmentation masks. We first select images from SA-1B in which at least one annotated instance occupies a moderate fraction of the image (roughly around $10\%$ to $20\%$ of the image area). For each selected image, we choose a single target instance and treat its binary mask as the GT $M_{\mathrm{gt}}$, making image–mask pairs $(I, M_{\mathrm{gt}})$ and associated geometric annotations, which later serve as GT references for both mask evaluation and synthetic trajectory construction. Then we construct a natural language problem $Q$ describing the target object. Concretely, we employ strong proprietary vision–language models (practically, we used Gemini2.5-Pro (Google, 2025) and OpenAI-o3 (OpenAI, 2025)) to generate problem in the style of reasoning segmentation according to $(I, M_{\mathrm{gt}})$.

## B.2. Multi-turn Tool Interaction for Trajectory Synthesis

After problem generation, we synthesize full multi-round trajectories with the seed pair $(I, Q)$ by letting a model with the ability of pixel-understanding interact with the same tool-augmented environment as RSAgent. Concretely, we place Qwen2.5VL-72B-Instruct (Bai et al., 2025) in our visual tool environment, which exposes the view manipulation and segmentation tools described in Section 3.2. Our goal is to obtain multi-round segmentation trajectories that closely match the interaction protocol of RSAgent at test time: at the beginning of each episode, the Qwen-VL model receives the original image and the problem $Q$; at each subsequent step, it observes the updated visual context (the base image together with a small pool of overlays summarizing previous masks) and the accumulated textual conversation, and produces the next reasoning segment and tool invocations. The resulting interaction yields a full pixel-space reasoning trajectory with multi-turn tool calls and corresponding visual states, closely aligned with the Markov decision process used later for reinforcement learning. The overall pipeline is illustrated in Figure 4.

To ensure that the synthetic trajectories provide reliable supervision, we evaluate the final prediction against the $M_{\mathrm{gt}}$. We select the cold-start data with two hard principles: (i) the IoU of the final predicted mask's and $M_{\mathrm{gt}}$ can't be lower than 0.9, and (ii) the number of reasoning turns shouldn't surpass 8. The former principle not only validates the accuracy of the problem, but also demonstrates the multi-turn interleaved visual and textual reasoning process's rightness. Unlike some previous "think with image" work (Wang et al., 2025b; Hong et al., 2025; Jiang et al., 2025; Zheng et al., 2025), which make processive VQA tasks and don't provide "final-answer" form of observation in middle turns, every tool call of SAM2 will produce a final result in history pool. Therefore, although some trajectory's final answer is correct, the actual reasoning process may be unreliable due to overly long contexts and compounding errors / hallucinated rationales.

In complex scenes where the agent cannot backtrack or undo previous operations, any suboptimal decision at an intermediate step may propagate and compound, adversely affecting subsequent predictions and ultimately degrading the final segmentation quality. In addition to the high quality trajectories selected under the above criteria, we also incorporate a small supplemental set of trajectories that exhibit a modest number of tool invocations and a low final mask IoU, but for which a correct mask is produced by SAM2 at some intermediate reasoning step. After manually verifying the correctness of the corresponding problem $Q$, we revise the terminal supervision to align with this correct intermediate mask and discard the most misleading portions of the erroneous reasoning trace, and then add these curated trajectories to the cold-start dataset.

## B.3. Analysis of our Generated Dataset

As shown in Table 7, our $D_{\mathrm{SFT}}$ completely follow the data generation pipeline, consisting of 5069 reasoning trajectories. We set the max number of tool invocations as 6, but the average turn of correct trajectories differs.

# C. Reward Design in RL Training

We formulate tool-augmented segmentation as a finite-horizon interactive episode. At each step $t$, the policy either invokes a tool (zoom/rotate/segment) or outputs a final answer in the required JSON schema. Our overall reward is decomposed into three components: (i) **Final-answer reward** $R_{\mathrm{final}}$ for final mask quality, (ii) **Format reward** $R_{\mathrm{format}}$ for valid `<think>`/

*Table 7.* Statistics of our cold-start SFT dataset $D_{\mathrm{SFT}}$. We report dataset scale, the distribution of tool-invocation rounds (with $T_{\max} = 6$), tool-type usage, target object category statistics, and the length statistics of questions.

| Cold-start SFT dataset ($D_{\mathrm{SFT}}$) statistics | | | |
|---|---|---:|---|
| **Panel A. Dataset scale & trajectory-level tool usage** | | | |
| Total trajectories | $\|D_{\mathrm{SFT}}\|$ | 5069 | reasoning trajectories generated by our pipeline |
| Max tool invocations | $T_{\max}$ | 6 | upper bound during generation |
| Avg. turns (all) | $\mathbb{E}[T]$ | 3.57 | average tool-invocation rounds per trajectory |
| **Panel B. Distribution of tool-invocation rounds ($T \in [0, 6]$)** | | | |
| $T = 0$ | no tool call | 7.1% | |
| $T = 1$ | 1 round | 14.3% | |
| $T = 2$ | 2 rounds | 19.9% | |
| $T = 3$ | 3 rounds | 20.6% | |
| $T = 4$ | 4 rounds | 26.8% | |
| $T = 5$ | 5 rounds | 7.1% | |
| $T = 6$ | 6 rounds (hit cap) | 4.2% | |
| **Panel C. Tool type statistics (trajectory-level and/or call-level)** | | | |
| Trajectories using Zoom-in | ratio | 43.8% | at least once in a trajectory |
| Trajectories using Rotate | ratio | 8.3% | |
| Trajectories using Segment | ratio | 90.9% | |
| Total tool calls | $\sum T$ | 18351 | total calls across all trajectories |
| Zoom-in calls | ratio | 37.7% | call-level distribution |
| Rotate calls | ratio | 8.4% | |
| Segment calls | ratio | 90.9% | |
| **Panel D. Target object category statistics** | | | |
| Top-1 category | architecture | 23.7% | |
| Top-2 category | animal or plant | 19.3% | |
| Top-3 category | natural object | 15.6% | |
| Top-4 category | industrial product or crafts | 13.1% | |
| Top-5 category | human body or clothes | 10.5% | |
| Long-tail (others) | count / ratio | 17.8% | categories outside top-5 |
| **Panel E. Question length statistics** | | | |
| Question tokens (avg / median) | tokens | 19.3 / 20 | tokenized by *(Qwen-2.5-VL-7B tokenizer)* |
| Question tokens (p25 / p75) | tokens | 10 / 28 | |
| Short question | $\leq \tau_{\mathrm{q}}$ | 37.1% | threshold $\tau_{\mathrm{q}} = 15$ tokens |
| Long question | $> \tau_{\mathrm{q}}$ | 62.9% | |

<tool_call> / <answer> blocks, and (iii) **Process reward** $R_{\mathrm{process}}$ for step-wise shaping along the trajectory.

## C.1. Process Reward $R_{\mathrm{process}}$

Following prior findings that step-level (process-based) supervision provides richer signals and eases long-horizon credit assignment compared to outcome-only feedback, we incorporate step-wise IoU-improvement shaping as a process reward to guide tool-use trajectories (Lightman et al., 2023; Uesato et al., 2022). Let $\mathrm{IoU}_t$ denote the foreground IoU (thresholded at $0.5$) between the current best predicted mask and the ground-truth mask after executing the $t$-th turn of tool call, and define $\Delta_t = \mathrm{IoU}_t - \mathrm{IoU}_{t-1}$. We use step-wise shaping to ease credit assignment:

$$r_t = \lambda_\Delta \, \mathrm{clip}(\Delta_t, -0.1, 0.5) + \lambda_{\mathrm{best}} \, \max(0, \mathrm{IoU}_t - \mathrm{IoU}_{t-1}^*) - c(a_t) - \lambda_{\mathrm{inv}} \cdot \mathbb{1}[\text{invalid}] + r_t^{\mathrm{pt}}, \tag{7}$$

where $\mathrm{IoU}_{t-1}^* = \max_{k \leq t-1} \mathrm{IoU}_k$ is the best-so-far IoU. We assign a tool-dependent call cost $c(a_t) = \lambda_{\mathrm{cost}} \cdot \kappa(a_t)$ with multipliers $\kappa(\cdot)$ (e.g., higher for segmentation of 2.5 than for geometric transforms of 1) to discourage excessive tool usage. We adopt $\lambda_\delta$ as 1, $\lambda_{best}$ as 0.5, $\lambda_{inv}$ as 1. Then the final $R_{\mathrm{process}}$ is calculated as:

$$R_{\mathrm{process}} = \eta \sum_{t=1}^{T} r_t, \tag{8}$$

where we default $\eta$ as 1 during training.

**Point-level sparsity/novelty (within $R_{\text{process}}$).** When invoking the segmentation tool, we additionally encourage *novel* point prompts while penalizing redundant clicks:

$$r_t^{\text{pt}} = \mathbb{1}[\Delta_t > \epsilon]\Big(\rho \cdot N_{\text{new}} - \beta_{\text{pt}} \cdot \text{Redund}\Big), \tag{9}$$

where $N_{\text{new}}$ counts points whose distance to the historical point set is at least $d_{\text{min}}$ of 8, and $\text{Redund}$ accumulates a normalized penalty for points closer than $d_{\text{min}}$, encouraging sparsity and reducing repeated clicks.

### C.2. Format Reward $R_{\text{format}}$

We reward syntactically valid, schema-compliant `<answer>` and `<tool_call>` blocks, and penalize unparsable outputs and invalid tool invocations. Concretely, $R_{\text{format}}$ includes (i) a small bonus for parsable, schema-compliant JSON outputs, and (ii) penalties for invalid tool calls or unparsable generations .

### C.3. Final-answer Reward $R_{\text{final}}$

Given the final `<answer>`, we run the segmentation model for each predicted item to obtain masks, and score them against the GT instance set using Hungarian matching. We define

$$R_{\text{final}} = \overline{\text{IoU}}_{\text{match}} + 0.5 \cdot \text{IoU}_{\text{box}}, \tag{10}$$

where $\overline{\text{IoU}}_{\text{match}}$ is the matched mean IoU over GT connected components, and $\text{IoU}_{\text{box}}$ is the IoU between the union bounding box of predicted masks and the GT union box. This term directly encourages high quality final masks via both mask IoU and bounding-box IoU.

**Episode return.** We aggregate the above components as Equation. 3 where $\beta$ controls the strength of dense process shaping relative to the final outcome. We adopt $\alpha$ of 1, $\beta$ of 0.5, $\gamma$ of 0.2 as our training parameter.

## D. Additional Description of Experiments

### D.1. Baseline Methods

We compare RSAgent with four families of representative methods on both RES and ReasonSeg. (*1*) *Discriminative vision–language segmentors without LLM-based controllers.* This group covers classical RES/RIS networks and open-vocabulary segmentors that rely on task-specific cross-modal encoders or CLIP-style backbones, but do not use a generative LLM/VLM as the decision making controller(e.g., LAVT (Yang et al., 2022), ReLA (Liu et al., 2023),GLEE (Wu et al., 2024), DETRIS (Huang et al., 2025a), UniLSeg (Liu et al., 2024) and EVF-SAM (Zhang et al., 2024) on RefCOCO-style benchmarks, as well as SEEM (Zou et al., 2023), Grounded-SAM (Ren et al., 2024a) and OVSeg (Liang et al., 2023) on ReasonSeg-style evaluations). (*2*) *Large (multimodal) models for single-pass segmentation without explicit reasoning traces.* These methods leverage large multimodal backbones and predict masks in a mostly one-shot manner, without exposing step-by-step textual reasoning during inference. Representative examples include GLaMM (Rasheed et al., 2024), GSVA (Xia et al., 2024), SAM4MLLM (Chen et al., 2024), PixelLM (Ren et al., 2024b), and RICE (Xie et al., 2025) on RES, together with early MLLM-based reasoning-segmentation baselines such as LISA (Lai et al., 2024), Hyperseg (Nirkin et al., 2021), X-SAM (Wang et al., 2025a). (*3*) *Explicit chain-of-thought (CoT) or reasoning-chain guided segmentation.* This family explicitly produces multi-step textual explanations or structured reasoning traces (often followed by localization/mask prompting) before committing to a mask. Typical examples include RSVP (Lu et al., 2025), as well as RL-enhanced reasoning-chain approaches such as Seg-Zero (Liu et al., 2025a), Seg-R1 (You & Wu, 2025) and SAM-R1 (Huang et al., 2025b). (*4*) *Multi-round tool-calling agents with iterative refinement.* Different from one-shot prediction or single-pass prompt generation, these agent-style frameworks iteratively call external segmentation tools, inspect intermediate masks, and update prompts over multiple rounds. A representative example is SAM3 Agent (Carion et al., 2025), which uses an MLLM to repeatedly propose prompts for SAM3 and refine them based on tool feedback. Together, these four groups cover non-LLM discriminative architectures, one-shot MLLM segmentors, explicit CoT-based reasoning segmentation, and multi-round tool-using agents, providing a comprehensive comparison ground for RSAgent.

*Table 8.* Controlled comparison between single-pass post-training and multi-turn post-training. All post-trained variants use the same synthetic supervision pipeline whenever applicable.

| Method | ReasonSeg test gIoU | ReasonSeg test cIoU | RES testA gIoU | RES testA cIoU |
|---|---|---|---|---|
| Base model (single-pass) | 50.1 | 43.9 | 64.4 | 64.9 |
| Single-pass + same synthetic SFT pipeline | 57.5 | 52.2 | 72.7 | 73.5 |
| Single-pass + same synthetic RL pipeline | 61.3 | 53.1 | 69.6 | 70.1 |
| RSAgent (multi-turn, SFT only) | 61.2 | 55.4 | 73.3 | 73.9 |
| RSAgent (multi-turn, RL only) | 62.0 | 54.3 | 76.1 | 77.3 |
| RSAgent (multi-turn, SFT + RL) | **66.5** | **57.9** | **81.3** | **81.8** |

*Table 9.* Additional evaluation on MESS under the FPSS protocol. MESS is used as a complementary cross-domain semantic segmentation benchmark rather than the primary text-guided segmentation setting.

| Model | General | Earth | Medical | Engineering | Agriculture | Avg. |
|---|---|---|---|---|---|---|
| LISA | 57.0 | 47.6 | 31.6 | 12.7 | 63.9 | 42.6 |
| RSAgent | **59.6** | **49.1** | **34.2** | **16.4** | **66.7** | **45.0** |

## D.2. Additional experiments

**Controlled comparison with single-pass post-training.** To better disentangle the effect of multi-turn interaction from that of additional synthetic supervision, we conduct a controlled comparison against stronger single-pass baselines. Specifically, we train a single-pass model using the same synthetic cold-start SFT pipeline as RSAgent, and further apply the same RL post-training recipe to obtain a single-pass post-trained baseline. This comparison allows us to evaluate whether the performance gain mainly comes from the agentic multi-turn formulation or merely from additional SA-1B-derived supervision and stronger post-training signals.

As shown in Table 8, the same synthetic SFT pipeline substantially improves the single-pass baseline, indicating that the synthetic cold-start data is indeed beneficial. However, the best single-pass post-trained model still remains clearly below RSAgent with multi-turn SFT and RL, with gaps of $5.2$ gIoU / $4.8$ cIoU on ReasonSeg test and $8.6$ gIoU / $8.3$ cIoU on RES testA. Moreover, under comparable post-training settings, the multi-turn variants generally outperform their single-pass counterparts, suggesting that the gains of RSAgent cannot be attributed solely to additional synthetic supervision, recipe complexity, or stronger training signals. Instead, the results support our central claim that iterative observe–reason–act interaction provides a material benefit for text-guided segmentation, especially when initial localization is ambiguous or requires visual refinement.

**Additional evaluation.** We report results on MESS (Blumenstiel et al., 2023), a multi-domain benchmark originally designed for zero-shot semantic segmentation. We note that MESS is not the exact same task setting as the text-guided segmentation problem studied in this paper, since it focuses on semantic segmentation across multiple visual domains rather than instance-level or reasoning-based text-guided segmentation. Nevertheless, it provides a useful stress test for evaluating whether RSAgent can generalize beyond the standard RES and reasoning segmentation benchmarks.

Following the few-shot prompted semantic segmentation (FPSS) protocol (Avogaro et al., 2025), we evaluate RSAgent on MESS and compare it with LISA. As shown in Table 9, RSAgent improves the average score from $42.6$ to $45.0$, with consistent gains across all five domain groups. These results provide additional evidence that the proposed multi-turn reasoning-and-acting formulation transfers beyond the primary text-guided segmentation benchmarks, while we emphasize that MESS should be interpreted as a complementary cross-domain semantic segmentation evaluation rather than a direct replacement for RES or ReasonSeg.

*Table 10.* Cold-start SFT settings. We fine-tune Qwen2.5-VL-7B-Instruct with LoRA on our multi-round trajectories before RL.

| Cold-start SFT parameters | | |
| --- | --- | --- |
| Category | Hyperparameter | Value |
| Template | Prompt template | qwen2_vl |
| Data | Max sequence length | 32K |
| Vision | image_max_pixels | 2,250,000 |
| Method | Finetuning type | LoRA |
| Method | LoRA rank ($r$) | 16 |
| Method | LoRA alpha ($\alpha$) | 32 |
| Method | LoRA target | all |
| Optim | Optimizer | AdamW |
| Optim | Learning rate | $2 \times 10^{-5}$ |
| Optim | Epochs | 2 |
| Optim | LR scheduler | cosine |
| Optim | Warmup ratio | 0.1 |
| Precision | Mixed precision | BF16 |
| Batch | Effective batch size | 128 |
| Batch | Per-device train batch size | 4 |
| Batch | Gradient accumulation steps | 8 |
| System | Preprocess workers | 16 |
| System | Dataloader workers | 4 |

# E. Training Details

**Training parameters.** We list our cold-start SFT and RL training parameters as Table 10 and Table 11. We select AdamW (Loshchilov & Hutter, 2017) as the optimizer of cold-start SFT and RL.

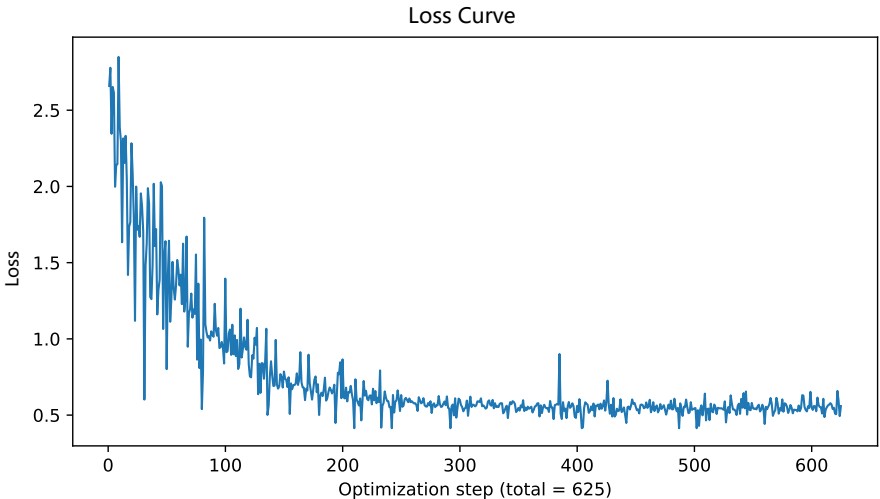

*Figure 6.* Loss curve during training.

**Loss and reward curves.** We visualize the changes in loss and rewards during the training process in Figure 6 and Figure 7. As shown in Figure 8, $R_{\text{format}}$ converges to 0.2 in few steps, the $R_{\text{final}}$ gradually increases over time, and $R_{\text{process}}$ increases relatively slower. This suggests that the format rewards initially dominates the optimization direction, leading to a decrease in response length during the initial training steps. However, as the format rewards converge and the accuracy rewards gradually increases, the model's completion length (e.g., the CoT reasoning process) begins to expand.

*Table 11.* RL training settings. We optimize RSAgent with GRPO in VERL after cold-start SFT.

| RL (VERL) training hyperparameters | | |
|---|---|---|
| Category | Hyperparameter | Value |
| Framework | RL framework | VERL |
| Algorithm | Advantage estimator | GRPO |
| Algorithm | Use kl loss | false |
| Data / Sampling | Batch size | 16 |
| Data / Sampling | Sampling number ($n$) | 4 |
| Data / Sampling | Seed | 42 |
| Data / Sampling | shuffle | true |
| Data / Sampling | episodes(size) | $1(10K)$ |
| Agent | Max tool-call turns | 8 |
| IO length (RL) | Max prompt length | $32K$ |
| IO length (RL) | Max response length | $2K$ |
| IO length (RL) | Max pixels | 1003520 |
| IO length (RL) | Min pixels | 3136 |
| Rollout | Temperature | 1.0 |
| Rollout | TP | 2 |
| Rollout | GPU memory utilization | 0.6 |
| Rollout | Limit images | 0 |
| Rollout | Enable chunked prefill | true |
| Optimization | Learning rate | $1 \times 10^{-6}$ |
| Optimization | Weight decay | $1 \times 10^{-2}$ |
| Optimization | Off load | true |

## F. More Visualization Examples

We display part of examples of RSAgent's performance on ReasonSeg test split. For observations of segmentation results in middle turns, we use the blue overlays to display, while for the answers we choose the green overlays for successful results and the red ones for failed results.

**Successful examples.** As shown in Figure 9 and Figure 10, we list several typical examples of RSAgent's multi-turn tool invocations in ReasonSeg test split, which strongly prove RSAgent's rethinking, relocalization and mask refinement ability.

**Failed cases.** We also list some unexpected cases RSAgent met in during evaluation in Figure 11. We contribute the majority of bad cases to three categories: (i) confusion of oversized object with the background, (ii) unsure of choice decision and (iii) for images with low resolution, similar features can easily mislead the model.

## G. Prompts

### G.1. Training Prompts

We list the system prompt, user prompt and turn prompt of RSAgent in Figure 12 and Figure 13, where system prompt is a complex and structured regulation and user prompt is just the description of the task object itself. The user prompt is used as the first turn, while the turn prompt is used as the following turns together with the original image and history observations.

**System prompt (protocol and tool specification).** The system prompt defines RSAgent as an interactive visual segmentation assistant and provides a complete tool contract in a machine-readable format (function signatures enclosed in `<tools>...</tools>`). Concretely, it exposes three primitives—`image_zoom_in_tool`, `image_rotate_tool`, and `sam2_segment`—together with strict argument schemas (e.g., `bbox_2d`, `point_coords`, `point_labels`, and optional `multimask_output` and `label`). Beyond capability exposure, the system prompt enforces hard constraints that make each interaction step well-defined: (i) the agent must invoke `sam2_segment` at least once before producing any final answer, (ii) all coordinates must be expressed in pixel space of the designated reference image, and (iii) tool invocations must be serialized as one or more `<tool_call>...</tool_call>` blocks with valid JSON arguments.

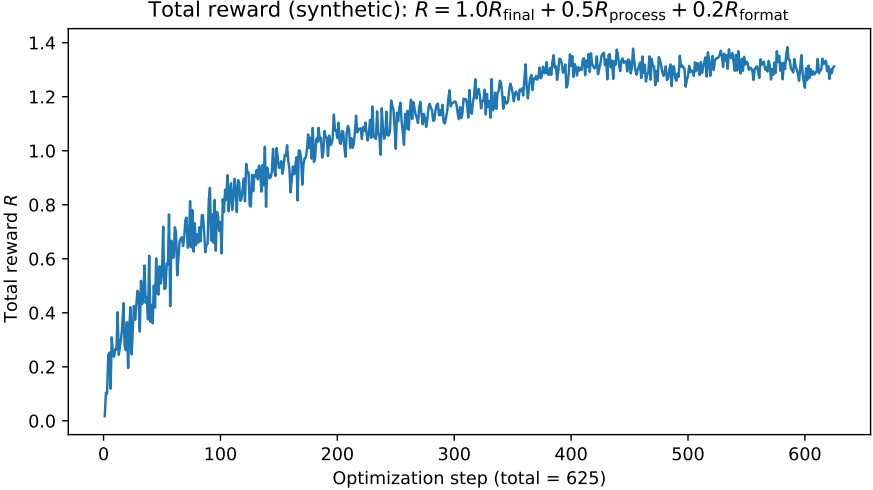

*Figure 7.* $R_{\text{total}}$ curve during training.

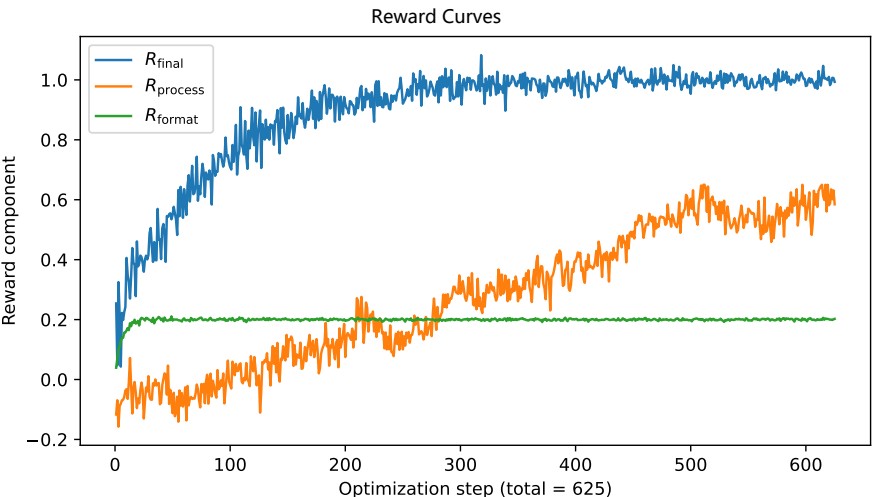

*Figure 8.* Curves of $R_{\text{format}}$, $R_{\text{final}}$ and $R_{\text{process}}$ during training.

These constraints encourage tool-grounded behavior, reduce premature termination, and improve trajectory reproducibility.

**Output schema and verifiability.** To make the agent output programmatically verifiable, the system prompt specifies a strict final-answer format: the final result must appear only inside an `<answer>...</answer>` block and must be a JSON array of one or more hypotheses. Each hypothesis may contain `bbox_2d` and/or `point_coords`, with `point_labels` aligned in length. Natural language inside `<answer>` is explicitly forbidden, and wrapping JSON in markdown fences is disallowed, preventing common parsing failures (e.g., mixed prose or JSON or trailing punctuation) and simplifying downstream reward computation from structured actions.

**User prompt (instance-level task).** In contrast to the protocol-heavy system prompt, the user prompt is intentionally minimal: it only contains the natural-language description of the target object. This separation avoids contaminating global rules with instance content and encourages the model to rely on iterative tool-mediated perception rather than memorizing task templates.

**Turn prompt (multi-turn state and execution invariants).** As shown in Figure 13, for all subsequent turns, we inject the turn prompt together with (a) tool responses (e.g., `<tool_response>...</tool_response>`) and (b) the updated image set. The turn prompt enforces a critical image convention: `Image[0]` is always the original full-resolution image, while additional images (e.g., `Image[1]`) are low-resolution historical previews used only for comparison. Accordingly, it requires that all coordinates be specified in the pixel space of `Image[0]` and prohibits placing points/boxes on preview images, which prevents coordinate drift under zoom/rotation and guarantees that the final answer remains aligned with the original image coordinate system. Finally, the turn prompt standardizes the decision format: the model first reasons inside `<think>...</think>`, and then outputs either (i) continued tool calls, or (ii) a single final `<answer>...</answer>` that strictly follows the schema from the system message.

### G.2. Question Generation Prompt in our Data Pipeline

As shown in Figure 4, both question generation and multi-turn trajectory generation need prompts with precise description and rational regulations. As illustrated in Section 3.3, the prompts for trajectory generation keep the same as the training setting, so we only list our prompt for question generation in Figure 14. The prompt prohibits mentioning "masks", "bounding boxes", "segmentations", "pixels", or "annotations", and disallows wrapping the question in JSON, ensuring natural phrasing. It further requires the question to be concise (roughly 8–30 words) and focused on a single target. Mentioning the object category is allowed when necessary, but the query should still emphasize the object's role rather than a trivial instruction, improving diversity and difficulty while keeping the supervision interface clean.

## H. Limitation and Future Work

As illustrated in Section F, despite strong performance, RSAgent has several limitations. (i) *Limitation of grounding ability.* The base model's grounding ability restricts the model to recognize some objects which enjoy similar features with their background. (ii) *Efficiency.* Multi-round tool invocation increases latency and compute, and long-context conditioning can make deployment harder in resource-constrained or real-time settings. (iii) *Tool-dependence and ceiling.* Our agent ultimately inherits an upper bound from the frozen segmentation toolbox: when the segmentor fails on tiny or huge objects, heavy occlusion, unusual viewpoints or fine boundaries, multi-turn reasoning may still struggle to fully recover. (iv) *Scope of evaluation.* Current experiments focus on still-image RES or ReasonSeg benchmarks, leaving generalization to video, 3D or embodied settings, and safety-sensitive scenarios remains to be validated.

Future work could includes: integrating stronger and more diverse segmentors (e.g., SAM3 (Carion et al., 2025)) while keeping the agent policy unchanged, exploring the fusion of concept and vision according to both SAM2's and SAM3's feedback, along with uncertainty-aware early stopping and budgeted planning; building a more open and scalable data pipeline with human-in-the-loop verification and open source teachers; extending the toolbox to video or temporal tools and long-horizon tracking; and improving efficiency via action compression, caching, and lightweight on-device inference.

User: If someone in the picture wanted to enjoy a drink without having to pour it themselves, what object could they use to easily take small sips at a time?

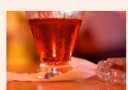

Turn 1 : <think>\nTo segment the glass as requested, I should focus on identifying the beverage container used for drinking sips. A shaker is typically not the answer given its purpose isn't for taking sips directly, unless it's filled with a ready-to-drink concoction.\n</think>\n<tool_call>\n{\"name\": \"sam2_segment\", \"arguments\": {\"bbox_2d\": [457, 5, 1986, 1737]}}\n</tool_call>\n<tool_call>\n{\"name\": \"image_zoom_in_tool\", \"arguments\": {\"bbox_2d\": [1478, 1318, 1851, 1630], \"label\": \"ice cubes nearby\"}}\n</tool_call>"

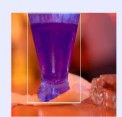
Obs1

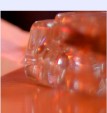
Obs2

Turn 2 : <think>The target object appears to be the drinking glass. Based on its design, it seems well-suited for taking small sips, as suggested by the question text. This leads me to conclude that the segmentation around the glass is adequate for the described task of enjoying a drink without having to pour oneself continuously. There's no need for further refinement.</think>\n<answer>\n[\n {\n \"bbox_2d\": [457, 5, 1986, 1737]\n }\n]\n</answer>

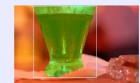
Final Answer

User: In order to take a clear and stable photo or video, what equipment in the picture can be used to hold the camera steady?

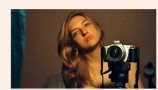

Turn 1 : <think>Given the request to identify equipment for holding a camera steady, let's start by segmentation the relevant areas in the image where such devices might appear. I'll begin by zooming in on what looks like a tripod or camera stabilizer, to refine the search area for these specific items.</think>\n<tool_call>\n{\"name\": \"image_zoom_in_tool\", \"arguments\": {\"bbox_2d\": [700, 400, 1700, 1000], \"label\": \"stabilization equipment\"

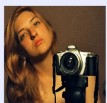
Obs1

Turn 2 : <think>I need to segment the camera tripod, as it seems like the stabilizing mechanism being referred to in the target description. The current bounding box appears to cover part of the camera, which is reasonable as we're zoomed in on a specific area.\n</think>\n\n<tool_call>\n{\"name\": \"sam2_segment\", \"arguments\": {\"bbox_2d\": [623, 580, 1588, 999], \"point_coords\": [], \"label\": \"a camera tripod stabilizing a camera\"}}\n</tool_call>

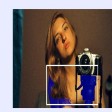
Obs2

Turn 3 : <think>The camera support system is visible down the tripod holding the camera, allowing it to be used to take a clear and stable photo. Based on the segment result from the image, I've prepared the final annotations that should capture the camera tripod area effectively for stability.</think>\n<answer>[\n {\n \"bbox_2d\": [1206, 659, 1545, 1003]\n\"point_coords\":[[1236 803]], \"point_labels\" :[0] }\n]</answer>

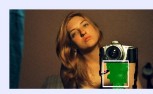
Final Answer

*Figure 9.* Part of successful results.

User: the area that displays the time 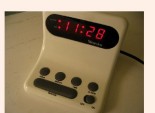

Turn 1 : <think>\nTo segment the area displaying the time, let's hypothesize its location by assuming it's typically found near the upper part of the screen in common clock interfaces, like a clock's face or date/time display. We might try zooming in around the top left corner to capture what seems to be a potential clock or time display area. \n</think>\n<tool_call>\n{\"name\": \"image_zoom_in_tool\", \"arguments\": {\"bbox_2d\": [10, 10, 430, 600], \"label\": \"potential area with a time display\"}}\n</tool_call>

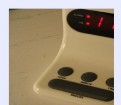
Obs1

Turn 2 : <think>The target area appears to be the digital clock display area. I will use a bounding box to identify it and check for any finer details such as the digits or labels to ensure I capture only the red time display.</think>\n<tool_call>\n{\"name\": \"sam2_segment\", \"arguments\": {\"bbox_2d\": [284, 42, 633, 161], \"label\": \"time display on the digital clock\"}}\n</tool_call>

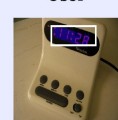
Obs2

Turn 3 : <think>I used the `sam2_segment` tool to segment the digital clock part with the time display '11:28'. The bounding box identified the area around the red alphanumeric display effectively capturing it without including other irrelevant parts like buttons, brand name or screen corners.</think>\n<answer>\n[\n {\n \"bbox_2d\": [239, 32, 622, 178]\n \"point_coords\":[[489 102], [603,48]], \"point_labels\" :[1,0] }\n]\n</answer>

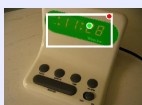
Final Answer

User: Creating visual representations of military equipment can be an interesting art form. What object in the picture could be a subject for a doodle or drawing related to the military? 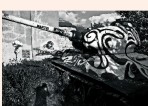

Turn 1 : <think>The image depicts an old tank covered in graffiti. This military equipment would indeed make a great subject for a doodle or drawing, focusing on the details and patterns created by the graffiti.</think><tool_call>{\"name\": \"sam2_segment\", \"arguments\": {\"bbox_2d\": [1305, 912, 3888, 2473]}}</tool_call>

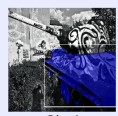
Obs1

Turn 2 : <think>I need to analyze which object in the image could be a subject for a military-themed doodle or drawing. However, I notice there isn't an image provided in context for me to locate the specific object visually. \n\nGiven this situation, it seems like I might need to use the `image_rotate_tool` to view the image from different angles to determine what part could be suitable as a military doodle. Since there's a lack of information about the image itself, I'll use the rotating tool first to see if anything stands out differently at rotated viewpoints.\n\nLet's attempt to rotate the image.</think>\n<tool_call>\n{\"name\":\"image_rotate_tool\", \"arguments\":{\"angle\":90}}\n</tool_call>

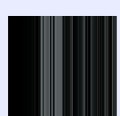
Obs2

Turn 3 : <think>The tank in the center is a clear candidate for the military equipment that could be a subject for a doodle or drawing, given its prominence in the image.</think>\n<tool_call>\n{\"name\": \"sam2_segment\", \"arguments\": {\"bbox_2d\": [1397, 267, 3886, 2597], \"label\": \"military tank\"}}\n</tool_call>

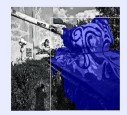
Obs3

Turn 4: <think>The segmentation step completed using SAM2 didn't  isolated the whole tank, which aligns with the request to visualize military equipment. This segmentation provides me understanding of the tank as instructed. Since there's a need for further refinement, I can now present the final answer.</think>\n<answer>\n[\n {\n \"bbox_2d\": [1388, 268, 3874, 2062]\n \"point_coords\":[[489 102], [603,48],[603,48]], \"point_labels\" : [1,1,0]}\n]\n</answer>

Final Answer

*Figure 10.* Part of successful results.

User: Looking at the image, it seems that the dog is engaged in a playful activity. What part of the dog's body is likely to be holding an object during this playful activity? 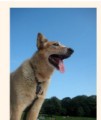

Turn 1 : <think>Based on the task, the primary focus should be on identifying the part of the dog's body likely to be holding an object, which could be its paws or mouth during a playful activity. However, as there is no direct mention of an object being held, we might need to zoom in on these parts to analyze further details. This suggests using 'image_zoom_in_tool' might be appropriate. Since there's no existing segmentation, starting with 'sam2_segment' would not make sense until I've visually assessed the image better.</think>\n<tool_call>\n\n{\"name\": \"image_zoom_in_tool\", \"arguments\": {\"bbox_2d\": [300, 250, 400, 370], \"label\": \"the dog's paw during play activity\"}}</tool_call>

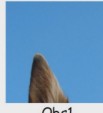
*Obs1*

Turn 2 : <think>Based on the current view, I notice the dog might have its mouth open, which could imply it's holding something. However, without additional context or a clearer view of the dog's paws, pinpointing exactly what part of its body holds an object isn't precise. A finer focus could help narrow down the area. Therefore, I'll zoom in further to capture a more detailed view of the dog's mouth area.</think>\n<tool_call>\n{\"name\": \"image_zoom_in_tool\", \"arguments\": {\"bbox_2d\": [326, 253, 690, 622], \"label\": \"closer view of the dog's mouth area\"}}</tool_call>

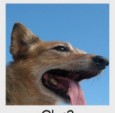
*Obs2*

Turn 3 : <think>Upon zooming in, I notice the dog's mouth is open, and there's some space in its mouth where an object could be held. I'll continue to explore refinement by possibly marking some points that seem promising around the lower jaw area.</think>\n<think<answer>\n{\"bbox_2d\": [393, 529, 625, 729]}\n</answer>

Confusion of oversized object with the background

*Final Answer*

User: the fried food 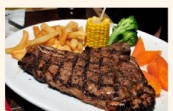

Turn 1 : <think>\nTo identify the \"fried food,\" which are likely the french fries, we need to segment the golden, crispy pieces at the top-left part of the image. These appear clearly separated from the other food items like steak, vegetables, etc., against a contrasting background. Using a segmented bounding box around these will target the fries accurately.\n</think>\n<tool_call>\n{\"name\": \"sam2_segment\", \"arguments\": {\"bbox_2d\": [0, 68, 425, 267]}}</tool_call>

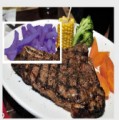
*Obs1*

Turn 2 : <think>It looks like there might be a misunderstanding. From the task description, \"the fried food\" refers to what I initially segmented as what appears to be the whole plate (Image[1]). This does not specifically align with something labeled as \"fried food\". However, I see now there's a need to clarify where the actual \"fried food\" was located, which seems to be just part of the original \"plate\" area.\n\nI should apply `box segmentation` on the location that includes the part resembling fries. Based on zoom, let's focus on the rightmost lower side that could indicate something that may resemble what people commonly refer to as \"fried food\", assuming this is clearly visible and not ambiguous due to the rest of the meal's components. This can then proceed with segmenting it further if needed.</think>\n<tool_call>\n{\"name\": \"sam2_segment\", \"arguments\": {\"bbox_2d\": [627, 257, 837, 391], \"label\": \"the fried food\"}}</tool_call>

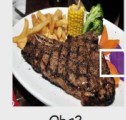
*Obs2*

Turn 3 : <think>Since the task is to find and segment \"the fried food\", I need to end by calling the `sam2_segment` tool with a rough bounding box around what appears to be a fried food item from the image.</think>\n<answer> {\"bbox_2d\": [0, 60, 820, 402]}\n</answer>

Unsure of choice decision

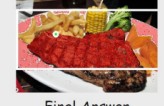
*Final Answer*

User: If the character in the picture is a wizard from the Harry Potter world, what would he ride on to fly in the air? 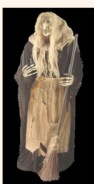

Turn 1 : <think>\nTo identify the object that represents what the wizard would ride in the Harry Potter world to fly, let's focus on the broomstick that the character is holding. Given the context, this should be the target object to segment correctly. We need to carefully locate the broomstick since it represents the primary means of transportation.\n</think>\n<tool_call>\n{\"name\": \"sam2_segment\", \"arguments\": {\"bbox_2d\": [96, 69, 210, 423]}}</tool_call>

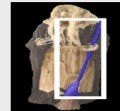
*Obs1*

Turn 2 : <think>The sam2_segment function provides a reasonable segmentation around the broomstick, which aligns with the description of a wizard's broom used for flying. It seems the initial segmentation is accurate based on the task definition.</think>\n<answer>\n[\n {\n \"bbox_2d\": [103, 99, 207, 451], }\n]\n</answer>

For images with low resolution, similar features can easily mislead the model

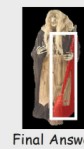
*Final Answer*

*Figure 11.* Part of failed results.

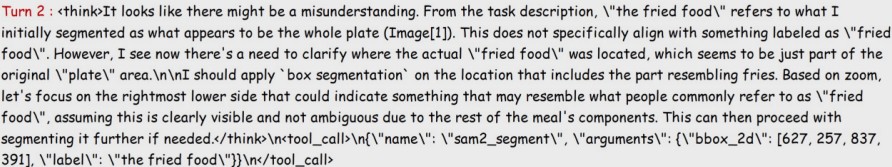
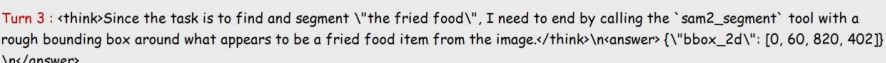
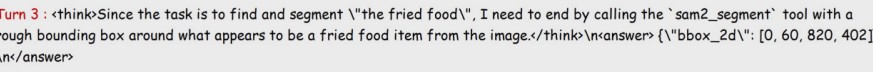

**System Prompt and User Prompt**

SYSTEM_PROMPT = """You are a helpful assistant specialized in interactive visual segmentation with tools.
Your goal is to use the tools to locate the object in the image that best matches the task description and provide the final answer that can accurately segmentation the task object.

# Tools
You are provided with function signatures within <tools></tools> XML tags:
<tools>
{"type":"function",
 "function":{"name":"image_zoom_in_tool",
            "description":"Zoom in on a specific region of the current image by cropping with a bounding box (bbox).",
            "parameters":{"type":"object",
                          "properties":{"bbox_2d":{"type":"array","items":{"type":"number"},"minItems":4,"maxItems":4,
                          "description":"Bounding box [x1, y1, x2, y2] in pixel coordinates of the CURRENT image."},
                          "label":{"type":"string","description":"Optional human-readable object hint."}},"required":["bbox_2d"]}}}
{"type":"function",
 "function":{"name":"image_rotate_tool",
            "description":"Rotate the current image by a specified angle (degrees).",
            "parameters":{"type":"object",
                          "properties":{"angle":{"type":"number",
                          "description":"Rotation angle in degrees. Positive = clockwise, negative = counterclockwise."}},"required":["angle"]}}}
{"type":"function",
 "function":{"name":"sam2_segment",
            "description":"Run SAM2 segmentation on the CURRENT image using an optional box and/or points (positive/negative). You may pass a bbox, or points+labels, or BOTH.",
            "parameters":{"type":"object",
                          "properties":{"bbox_2d":{"type":"array","items":{"type":"number"},"minItems":4,"maxItems":4,
                          "description":"Optional bounding box [x1, y1, x2, y2] in pixels."},"point_coords":{"type":"array","items":{"type":"array","items":
{"type":"number"},"minItems":2,"maxItems":2},"description":"Optional list of points [[x,y], ...] in pixels."},"point_labels":{"type":"array","items":{"type":"integer"},"description":"If
point_coords is provided, you MUST provide point_labels with the same length. Use 1 for positive and 0 for negative points."},"multimask_output":{"type":"boolean","description":"If
true, return multiple candidate masks; otherwise a single best mask."},"label":{"type":"string","description":"Optional human-readable object hint."}},"required":[]}}}
</tools>

# How to call a tool(HARD CONSTRAINTS)
You MUST CALL AT LEAST ONE "sam2_segment" TOOL BEFORE YOU EVER OUTPUT <answer>. In this turn, return one or more tool calls ONLY:
<tool_call>
{"name": <function-name>, "arguments": <args-json-object>}
</tool_call>

Rule(read carefully)):
- Coordinates are in pixel space of the CURRENT image.
- If you pass "point_coords", also pass "point_labels" of the same length (1=positive, 0=negative).
- You may call multiple tools in one turn.
- You'd better call "sam2_segment" at least once before giving any final <answer>.
- Do NOT wrap JSON in backticks; no comments or trailing commas.
- End your message with one or more <tool_call> blocks. Do NOT output <answer> in that turn.
- Maintain state across tool calls in the same turn: when refining, RE-USE and ACCUMULATE all previous "point_coords/point_labels" (and optional bbox) instead of replacing them.
- Add positive points (label=1) to major FN regions and negative points (label=0) to major FP/leakage regions near boundaries.
- Stopping:When you decide that the segmentation is good enough OR the point budget is reached, stop and output <answer>.
- If you zoom/rotate, remember coordinates are in the CURRENT image; FINAL <answer> must be in the ORIGINAL iordinate system.

# Final answer format (VERY IMPORTANT)
When you decide that the segmentation is good enough and you want to stop,
you MUST output the final result ONLY inside an <answer>...</answer> block,
and the content MUST be a JSON ARRAY (list) of one or more objects.

Each object describes ONE segmentation hypothesis and MUST have this structure:

<answer>
[
  {
    "bbox_2d": [x1, y1, x2, y2],
    "point_coords": [[x1_pos, y1_pos], [x2_pos, y2_pos], ...],
    "point_labels": [1, 1, 0, ...]
  }
]
</answer>

Rules for the final answer:
- The outermost structure inside <answer> MUST be a JSON list: [ {...}, {...}, ... ].
- "bbox_2d" is OPTIONAL but if present MUST be 4 numbers [x1, y1, x2, y2] in pixel coordinates of the ORIGINAL image (Image[0]).
- "point_coords" is OPTIONAL but if present MUST be a list of [x, y] points in pixel coordinates of the ORIGINAL image (Image[0]).
- "point_labels" is OPTIONAL but:
  - If "point_coords" is provided, you MUST provide "point_labels" with the SAME length.
  - Use 1 for positive points and 0 for negative points.
- You MAY omit "point_coords" and "point_labels" if you only want to use bbox_2d, but the JSON must remain valid.
- DO NOT include any natural language or comments inside <answer>, ONLY pure JSON.
- DO NOT wrap the JSON in ```json fences inside <answer>.
- All reasoning, explanations, and verbal description MUST go into <think>...</think>, NEVER inside <answer>.

**Example**:
<tool_call>
{"name":"sam2_segment","arguments":{"bbox_2d":[120,80,420,360],"point_coords":[[160,140],[260,210]],"point_labels":[1,0],"multimask_output":false, "label": "the apple on the desk"}}
</tool_call>
<tool_call>
{"name":"image_zoom_in_tool","arguments":{"bbox_2d":[150,120,260,240], "label": "the person under the tree"}}
</tool_call>
"""

USER_PROMPT = {Problem}

*Figure 12.* The system prompt and user prompt of RSAgent.

**Mid Turn Prompt**

TURN_PROMPT = """
Above are the tool responses (<tool_response>...) and updated image(s).

Image convention for this task:
- Image[0] is ALWAYS the ORIGINAL full-resolution image.
- Image[1:] are LOW-RES historical previews produced in previous steps for COMPARISON only.
- All coordinates (bbox/points) MUST be specified in the pixel space of Image[0].
- NEVER place points or boxes on preview images (Image[1:]); use them only to decide your next action.

Think first inside <think>...</think>, then EITHER continue calling tools OR provide the final answer.

If you choose to call tools:
- Output one or more <tool_call>{...}</tool_call>, each containing a valid JSON object with optional "bbox_2d", "point_coords", "point_labels", "multimask_output", "label".
- If you pass "point_coords", you MUST pass "point_labels" with the same length (1=positive, 0=negative).

If you choose to provide the final answer:
- You MUST follow the FINAL ANSWER JSON FORMAT described in the system message:
 <answer>
 [
  {
    "bbox_2d": [x1, y1, x2, y2],
    "point_coords": [[x_pos1, y_pos1], [x_pos2, y_pos2], ...],
    "point_labels": [1, 1, 0, ...]
  }
 ]
 </answer>
- NO natural language inside <answer>; only pure JSON.

Format STRICTLY as:
<think>...</think>
EITHER one or more <tool_call>{...}</tool_call>
OR a single <answer>...</answer>
"""

*Figure 13.* The turn prompt of RSAgent.

**Reasoning Problem Generation Prompt**

REASONING_PROBLEM_PROMPT="""
You are creating training data for a visual reasoning segmentation benchmark.

You will be given:
- a natural-language description of a target region/object in an image (this is the object covered by the mask),
- and possibly some brief context about the whole scene.

Your task is to write ONE short English "problem" that refers to this target object,
in a style similar to the following examples:

- "the equipment that can assist disabled individuals in mobility"
- "It is too hard for the man to lift up, can you segment something that he should drop?"
- "If someone in the picture wanted to enjoy a drink without having to pour it themselves, what object could they use to easily take small sips at a time?"
- "something that eats leaves"
- "the region exhibiting unusual color"
- "Jewelry is often made with precious gemstones, such as emeralds and turquoise. What piece of jewelry in the picture may contain these gemstones?"
- "the object that both teams are fighting for in a game of American football"

Style and content requirements:
1. The problem must point to the masked target using its function, role, effect, or typical usage,
  or its relation to other entities, not just its color or position.
2. It should require some mild reasoning or world knowledge, for example by:
   - describing what people want to achieve and asking which object they would use,
   - mentioning the rules of a game or a typical situation,
   - or focusing on what the object does (protects, supports, contains, transports, etc.).
3. It can be either:
   - a short noun phrase (starting with e.g. "something that...", "the object that...", "the region that..."), OR
   - one or two concise sentences that end with a question.
4. It is allowed to mention the object category name if needed, but you should still emphasize
  its function or role, not just say "segment the [object name]".
  For example:
   - OK: "the part of the shoe that provides extra height"
   - OK: "What device in the picture would someone use to take a close-up photograph?"
5. Do NOT mention masks, bounding boxes, segmentations, pixels, or anything about datasets or annotations.
  The problem should sound like a natural question or phrase that a human might ask. Do NOT wrap it in JSON or quotes.
6. The problem should be reasonably short (around 8–30 words) and focused on ONE target object/region."""

*Figure 14.* The question generation prompt of RSAgent.

