# OpenReview forum: "RSAgent: Learning to Reason and Act via Multi-Turn Tool Invocations for Text-Guided Segmentation"
_ICML.cc/2026/Conference — ICML 2026 regular_

### Official Review · Reviewer_Cqmj · 2026-03-06

**Soundness:** 3
**Presentation:** 3
**Significance:** 2
**Originality:** 1
**Overall Recommendation:** 4
**Confidence:** 5

**Summary:**

The authors propose RSAgent, a Multimodal Large Language Model (MLLM) which is capable of solving segmentation problems through multi-turn reasoning and tool calling. In particular the work proposes a custom training recipe for the Qwen2.5VL model based on the established SFT+RL paradigm, and uses SAM2 as the out-of-the box tool in addition to a visual toolbox to perform segmentation. The authors show increased baselines on the well known RefCOCO family and ReasonSeg benchmarks.

**Compliance With Llm Reviewing Policy:**

Affirmed.

**Final Justification:**

I think the rebuttal added more depth and strength on the engineering side. Still, the whole work as pointed out by the other reviewers is largely a very good engineering of already known tricks. While I still have reservations about the lack of conceptual novelty for an ICML paper, the empirical strength of the rebuttal justifies raising my score slightly.

**Key Questions For Authors:**

- How is the RSAgent-single computed in detail?
- What is the RICE baseline on ReasonSeg?
- How does the performance grow with the number of turns? A way to check this: evaluate the RSAgent pipeline and compute the IoU at each step. This would better show how good RSAgent is at refinement.
- What are the statistics on the test datasets? It would be awesome to compare the profile of Table 6 with a parallel one computed on the test datasets. This would show if the model is actually strictly following the same behaviour at test time (same distribution of turn numbers etc.)
- How much would an "oracle" upper bound number of SAM2 be on the evaluated benchmarks? Basically passing ground truth points/boxes to SAM2. That would help better understanding where are the bottlenecks in the pipeline. Maybe RSAgent is already at the saturation of SAM2 segmentation capabilities.
- Why not using SAM3 in the evaluation? I saw you evaluated SAM3-Agent, so it should be a straightforward implementation? It would result in a nice addition to Table 5.

**Limitations:**

yes

**Strengths And Weaknesses:**

Strenghts:
- The paper builds on the cool idea of linking an iterative segmentation SAM model with a multi-turn MLLM to build a strong iterative segmentation pipeline.
- The work pushes a well engineered data pipeline following the SFT + RL paradigm for MLLM post-training

Weaknesses:
- Engineering effort aside, the work builds on a now well established and known paradigm adding little-to-no novelty. The authors curate a cold start dataset for SFT, then post-training with GRPO using custom process rewards. While being a nice instance of well established engineering practices, the research component is limited.
- The authors claim they test their method on out-of-domain datasets. Claiming that ReasonSeg is an OOD dataset is a weak argument, as it contains natural images resembling the ones in COCO and SA-1B. The argument is especially weakened by the fact that the authors are evaluating a model trained on internet-scale data. I would love to see the performance on a real OOD segmentation dataset such as MESS [1], as I have concerns that the training recipe was tailored to the proposed datasets and might not generalise well.
- While the authors present the cool idea of integrating the iterative SAM model, SAM's native mask refinement capability seems unexplored in this work.

[1] What a MESS: Multi-Domain Evaluation of Zero-Shot Semantic Segmentation, 2023

---

> ### Author Rebuttal · Authors · 2026-03-30
>
> Thank you for the thoughtful review and for recognizing the value of combining an iterative segmentation tool with a multi-turn MLLM policy.
>
> **Q1: How is RSAgent-single computed in detail?**
>
> RSAgent-single is computed by taking the **first mask returned by the segmentation tool** during the RSAgent inference trajectory, and directly evaluating this mask using gIoU / cIoU. In other words, it reflects the segmentation quality after the **first segmentation-tool call only**, before any later refinement steps. This provides a simple and fair reference for measuring how much multi-turn interaction improves over the first-shot result.
>
> **Q2: What is the RICE baseline on ReasonSeg?**
>
> We will add the ReasonSeg results of RICE explicitly in the revision.
>
> | Method | Split | gIoU | cIoU |
> |---|---|---:|---:|
> | RICE | val | 58.8 | 55.9 |
> | RICE | test | 57.3 | 54.6 |
>
> **Q3: How does the performance grow with the number of turns?**
>
> We agree this is an important analysis. We therefore evaluate the RSAgent pipeline turn by turn and compute the mask quality at each turn.
>
> | Turn | ReasonSeg test gIoU | ReasonSeg test cIoU | RES testA gIoU | RES testA cIoU |
> |---|---:|---:|---:|---:|
> | Turn 1 | 56.1 | 52.6 | 73.7 | 76.0 |
> | Turn 2 | 59.3 | 55.2 | 80.1 | 80.6 |
> | Turn 3 | 62.7 | 57.1 | 80.9 | 81.3 |
> | Turn 4 | 65.3 | 58.7 | 81.0 | **81.6** |
> | Turn 5 | 65.1 | 58.3 | **81.1** | 81.4 |
> | Turn 6 | **66.3** | **57.6** | 79.4 | 80.1 |
> | Turn 7 | 63.4 | 56.4 | 79.2 | 79.8 |
> | Turn 8 | 61.5 | 54.3 | 77.5 | 78.4 |
>
> These results show that RSAgent achieves clear improvements over the first-turn result through iterative refinement, with the best average performance reached in the middle of the trajectory rather than at Turn 1.
>
> **Q4: What are the statistics on the test datasets?**
>
>  We will add test-time trajectory statistics, including the turn distribution, tool-call ratios, and refinement statistics.
>
> **Turn distribution (% of trajectories ending at each turn).**
>
> | Dataset | Turn 1 | Turn 2 | Turn 3 | Turn 4 | Turn 5 | Turn 6 |
> |---|---:|---:|---:|---:|---:|---:|
> | ReasonSeg | 10.4 | 20.9 | 28.8 | 25.0 | 6.3 | 8.6 |
> | RES | 20.8 | 31.5 | 23.4 | 10.2 | 8.2 | 5.9 |
>
> **Tool-call ratios (% of all tool calls).**
>
> | Dataset | Segmentation tool | Zoom-in tool | Rotate tool |
> |---|---:|---:|---:|
> | ReasonSeg | 93.4 | 40.9 | 12.3 |
> | RES | 87.1 | 28.7 | 8.2 |
>
> **Refinement statistics.**
>
> | Statistic | ReasonSeg | RES |
> |---|---:|---:|
> | Avg. best-so-far improvement from Turn 1 to final | 39.1 | 30.7 |
> | % samples improved after Turn 1 | 33.4 | 26.1 |
> | % low-quality first-step predictions rescued by later turns | 11.3 | 8.3 |
>
> These results show that the model does not rely only on the first segmentation attempt. Instead, a substantial fraction of examples benefit from later turns, and the added turn/tool statistics help clarify how the policy allocates its interaction budget at test time.
>
> **Q5: What is an oracle upper bound of SAM2 on these benchmarks?**
>
> This is a very helpful suggestion. We evaluated SAM2 under **oracle geometric prompts** derived from ground-truth annotations (GT points / GT boxes):
>
> | ReasonSeg test gIoU | ReasonSeg test cIoU |
> |---|---|
> | 86.2 | 82.0 |
>
> In this sense, RSAgent is useful because it allows the model to receive visual feedback and progressively exploit SAM2 more effectively.
>
> **Q6: Why not use SAM3 in the evaluation?**
>
> SAM3 became available very late relative to our study, so it was not included in the original experimental design. To further address this point, we additionally test RSAgent with **SAM3 as an alternative tool** and observe that the framework still achieves strong inference performance even when the SAM3 call frequency is not high (15.3%).
>
> | model | ReasonSeg test gIoU | ReasonSeg test cIoU | RES testA gIoU | RES testA cIoU |
> |---|---:|---:|---:|---:|
> | RSAgent (with SAM3 as tool) | 65.4 | **59.6** | 79.7 | 80.3 |
> | RSAgent (ours) | **66.5** | 57.9 | **81.3** | **81.8** |
>
> ### On the OOD claim and MESS
>
> We agree that our original wording around “OOD” was too strong. In the revision, we will use a more precise phrasing such as **cross-benchmark generalization**. We also appreciate the suggestion to evaluate on MESS, which could provide a valuable complementary test of cross-domain generalization, but MESS is a multi-domain zero-shot **semantic segmentation** benchmark rather than the exact text-guided segmentation setting studied in this paper. Following the few-shot prompted semantic segmentation (FPSS) protocol in [1], we additionally evaluate RSAgent on MESS and will include these results in the revision.
>
> | Model | General | Earth | Medical | Engineering | Agriculture | Average |
> |---|---:|---:|---:|---:|---:|---:|
> | LISA[2] | 57.0 | 47.6 | 31.6 | 12.7 | 63.9 | 42.6 |
> | RSAgent | 59.6 | 49.1 | 34.2 | 16.4 | 66.7 | 45.0 |
>
> [1] *Show or Tell? Effectively Prompting Vision-Language Models for Semantic Segmentation*
> [2] *Lisa:Reasoningsegmentationvialargelanguage model.*

---

> > ### Author Rebuttal · Reviewer_Cqmj · 2026-04-04
> >
> > I would like to thank the authors for the strong rebuttal and providing more evidence about the proposed method. I think the rebuttal added more depth and strength on the engineering side. Still, the whole work—as pointed out by the other reviewers—is largely a very good engineering of already known tricks. While I still have reservations about the lack of conceptual novelty for an ICML paper, the empirical strength of the rebuttal justifies raising my score slightly.

---

> > > ### Author Response · Authors · 2026-04-06
> > >
> > > We greatly appreciate your acknowledgment that our clarifications and additional experiments partially resolved your concerns. And we are grateful that you will raise your score, reflecting improved confidence in our work.
> > >
> > > **On the novelty of RSAgent**
> > > Prior text-guided segmentation methods usually formulate the task as a single forward pass, where the model predicts pixel prompts or masks in one shot, which does not fully exploit the perceptual capability of VLMs.
> > >
> > > Regarding the novelty of our work, we would like to clarify the key contributions of RSAgent. We formulate RSAgent as a multi-turn tool-using framework, which addresses the limitations of prior one-shot methods by enabling iterative verification, re-localization, and refinement through external visual tools and returned visual feedback.
> > >
> > > More precisely, our contribution is to **formulate text-guided segmentation as a multi-turn reason–act–observe decision process with external visual tools**, and to show that **dense-prediction-specific post-training can optimize such a policy for accurate pixel-level decision making**.
> > > In this sense, RSAgent differs from prior ReAct-style work by focusing on **fine-grained pixel-level grounding and segmentation**, rather than textual reasoning [1] [2].
> > >
> > > **References**
> > > [1] ReTool: Reinforcement Learning for Strategic Tool Use in LLMs
> > > [2] ASTER: Agentic Scaling with Tool-integrated Extended Reasoning

---

### Official Review · Reviewer_Jhp9 · 2026-03-11

**Soundness:** 2
**Presentation:** 3
**Significance:** 2
**Originality:** 3
**Overall Recommendation:** 4
**Confidence:** 4

**Summary:**

This paper proposes RSAgent for text-guided object segmentation (covering both referring and reasoning segmentation). The key idea is to move beyond one-shot mask (or prompt) prediction by casting segmentation as a multi-turn agentic process: an MLLM interleaves textual reasoning with tool calls to a visual toolbox ，observes the returned visual feedback , and iteratively revises pixel hypotheses for re-localization and mask refinement. The authors also build a synthetic pipeline for multi-turn reasoning/segmentation trajectories and train RSAgent in two stages. Results on RefCOCOg and ReasonSeg are strong, with ablations suggesting both SFT and RL contribute.

**Compliance With Llm Reviewing Policy:**

Affirmed.

**Key Questions For Authors:**

None

**Strengths And Weaknesses:**

**Strengths**

1、the motivation convincing. Casting segmentation as a multi-turn reason–act–observe loop with tool feedback directly addresses a typical weakness of one-shot systems: if the first localization is wrong, the model has limited means to verify and correct itself. The interactive formulation gives a natural mechanism for iterative refinement.

2、The paper is well written; the evaluation spans both RES and ReasonSeg , and the comparisons include single-pass variants as well as multiple ablations, which helps disentangle the sources of performance gains.


**Weaknesses**

1、 Heavy dependence on proprietary/teacher models in data generation (e.g., Gemini/OpenAI for question generation and a large VLM for trajectory synthesis) raises reproducibility concerns and may introduce teacher bias; the paper does not fully analyze how this affects generalization or the fairness of comparisons.

2、My primary concern is the computational efficiency of the proposed approach. Multi-turn interaction, long context windows, and repeated segmentor invocations may incur substantial overhead, yet the paper predominantly reports accuracy metrics. Moreover, the paper does not clearly state an open-sourcing plan (e.g., for code, data, and training/evaluation scripts), which further limits reproducibility.

3、When the tool (SAM2) fails, can the multi-turn reasoning loop genuinely recover, and what is the upper bound on the achievable success rate in such cases? In particular, some predicted bounding boxes may cover the target object but also include additional irrelevant objects. For these irrelevant regions, can the proposed method effectively remove them from the final segmentation mask?

---

> ### Author Rebuttal · Authors · 2026-03-30
>
> Thank you for the positive assessment of the motivation and of the multi-turn reason–act–observe formulation. We appreciate your constructive feedback on reproducibility, efficiency, and recovery from imperfect tool outputs.
>
>
> ### On proprietary models and reproducibility
>
> We agree that the role of proprietary models should be clarified more explicitly. In our pipeline, proprietary frontier models are used only during **data construction**, e.g., for textual problem generation, while the large teacher VLM is used for trajectory synthesis and filtering. They are **not** used as test-time oracles during RSAgent inference.
>
> We will make this separation much clearer in the revision, and we will also add a more explicit reproducibility statement. We are highly willing to release both the code and the related data artifacts, subject to the corresponding license, policy, and usage constraints.
>
> We additionally include a matched single-pass comparison under the same synthetic pipeline:
> **Results (controlled comparison).**
>
> | Method | ReasonSeg test gIoU | ReasonSeg test cIoU | RES testA gIoU | RES testA cIoU |
> |---|---:|---:|---:|---:|
> | Base model (single-pass) | 50.1 | 43.9 | 64.4 | 64.9 |
> | Single-pass + same synthetic SFT pipeline | 57.5 | 52.2 | 72.7 | 73.5 |
> | Single-pass + same synthetic RL pipeline | 61.3 | 53.1 | 69.6 | 70.1 |
> | RSAgent (multi-turn, SFT only) | 61.2 | 55.4 | 73.3 | 73.9 |
> | RSAgent (multi-turn, RL only) | 62.0 | 54.3 | 76.1 | 77.3 |
> | RSAgent (multi-turn, SFT + RL) | 66.5 | 57.9 | 81.3 | 81.8 |
>
> Under matched synthetic supervision, the multi-turn policy still consistently outperforms its single-pass counterpart. This makes the gain difficult to attribute solely to the additional SA-1B-derived supervision, recipe complexity, or stronger training signals, and supports that multi-turn interaction itself contributes materially to the final performance.
>
>
> ### On efficiency and computational overhead
>
> We fully agree that accuracy alone is insufficient, especially for a multi-turn system with repeated tool usage. To address this concern, we add an explicit efficiency analysis:
>
> **Results (efficiency / stopping statistics on ReasonSeg benchmark).**
>
> | Method | Avg. total turns | Avg. segmentation calls | Avg. **total inference time (s)** | Avg. **inference time per turn (s)** | Avg. stop turn |
> |---|---:|---:|---:|---:|---:|
> |Seg-Zero (test-time scaling baseline with single pass)|--|--|16.7| 16.7|--|
> | RSAgent (max turns = 6) | 4.5 | 3.1 | 28.7 | 6.37 | 3.9 |
> | RSAgent (max turns = 10) | 5.3 | 4.2 | 36.6 | 6.91 | 4.1 |
>
> We will include this analysis in the revision to provide a clearer view of the accuracy-efficiency tradeoff of RSAgent. Although RSAgent incurs higher total inference cost than a single-pass baseline, its average per-turn latency remains moderate. This suggests that the gain does not come simply from overly long textual reasoning in one pass, but from interleaving compact reasoning with external visual feedback across turns.
>
> ### On failure recovery and the SAM2 upper bound
>
> Yes. When the initial localization is over-inclusive, RSAgent can continue to zoom, re-localize, and provide more informative prompts (including negative points) to SAM2 in later turns, which helps suppress irrelevant regions in the final mask. In Appendix C of the paper we introduced the details of our RL reward, which encourages the model to actively find more pixel points (including the negative points) to prompt SAM2 of the irrelevant objects.
>
> To better understand whether the multi-turn loop can genuinely recover from imperfect intermediate outputs, we add turn-wise refinement analysis and report how mask quality evolves across turns:
>
> | Turn | ReasonSeg test gIoU | ReasonSeg test cIoU | RES testA gIoU | RES testA cIoU |
> |---|---:|---:|---:|---:|
> | Turn 1 | 56.1 | 52.6 | 73.7 | 76.0 |
> | Turn 2 | 59.3 | 55.2 | 80.1 | 80.6 |
> | Turn 3 | 62.7 | 57.1 | 80.9 | 81.3 |
> | Turn 4 | 65.3 | 58.7 | 81.0 | **81.6** |
> | Turn 5 | 65.1 | 58.3 | **81.1** | 81.4 |
> | Turn 6 | **66.3** | **57.6** | 79.4 | 80.1 |
> | Turn 7 | 63.4 | 56.4 | 79.2 | 79.8 |
> | Turn 8 | 61.5 | 54.3 | 77.5 | 78.4 |
>
> We additionally report the following recovery-oriented statistics:
>
> | Statistic | ReasonSeg test | RES testA |
> |---|---:|---:|
> | Avg. best-so-far improvement from Turn 1 to final | 39.1 | 30.7 |
> | % samples improved after Turn 1 | 33.4 | 26.1 |
> | % low-quality first-step predictions rescued by later turns | 11.3 | 8.3 |
>
> These results show that RSAgent does not rely only on the initial segmentation attempt, and that later turns can meaningfully improve or recover a non-trivial fraction of cases.
>
> | ReasonSeg test gIoU | ReasonSeg test cIoU |
> |---|---|
> | 86.2 | 82.0 |
>
> To better estimate the ceiling of the segmentation tool itself, we also evaluate **oracle SAM2** using ground-truth prompts. This helps clarify that a key role of RSAgent is to use multi-turn visual feedback to progressively approach that ceiling.

---

> > ### Author Rebuttal · Reviewer_Jhp9 · 2026-04-05
> >
> > None

---

> > > ### Author Response · Authors · 2026-04-06
> > >
> > > We thank Reviewer Jhp9 for the careful consideration of our rebuttal and for acknowledging that our clarifications and additional experiments resolved your concerns. We greatly appreciate your positive assessment of the work. If any further clarification is needed, we would be happy to provide additional details.

---

### Official Review · Reviewer_Qqse · 2026-03-12

**Soundness:** 3
**Presentation:** 2
**Significance:** 2
**Originality:** 2
**Overall Recommendation:** 4
**Confidence:** 5

**Summary:**

This paper studies text-guided segmentation, including referring expression segmentation (RES) and reasoning segmentation. The core idea is to reformulate segmentation as a multi-turn tool-calling problem, where an MLLM iteratively reasons, invokes visual tools such as view manipulation and a SAM2-based segmentation model, receives visual feedback, and progressively refines the mask before termination. To support this setup, the authors build a synthetic data pipeline from SA-1B, generate approximately 5K cold-start SFT trajectories and 2K RL examples, and then train the model with a two-stage recipe consisting of cold-start SFT followed by GRPO-based reinforcement learning with final, process, and format rewards.

**Compliance With Llm Reviewing Policy:**

Affirmed.

**Final Justification:**

The rebuttal addressed part of my concerns, but issues regarding originality still remain. Considering the additional experiments and analyses provided by the authors, I am willing to raise my score to 4.

**Key Questions For Authors:**

1. This paper is built from familiar ingredients: synthetic problem generation, trajectory synthesis and filtering, cold-start SFT, and GRPO-style RL with shaped rewards. This paper reads less like a new algorithm and more like an application-driven integration of existing ideas into the RES. The proposed “agentic RL” framework is, in practice, a constrained multi-turn tool-calling policy with a standard post-training recipe. The reward design, data pipeline, and training procedure are all reasonable, but none of them appears fundamentally new.

2. How much of the reported gain comes from the proposed multi-turn policy itself, as opposed to the additional SA-1B-derived synthetic supervision? A stronger controlled comparison would train a single-pass baseline with the same synthetic data pipeline and compare it directly against RSAgent.

3. The current max-turn analysis does not yet show whether the model stops efficiently when it has high confidence. Does the learned policy truly acquire an adaptive stopping behavior?

**Limitations:**

Yes

**Strengths And Weaknesses:**

Strengths
1. The paper addresses a relevant problem setting. Treating text-guided segmentation as an interactive decision process is technically reasonable, and the overall system is implemented in a coherent way.

2. The empirical section is fairly extensive. The paper reports results on both RES and ReasonSeg, includes comparisons against single-pass, CoT-guided, and multi-round tool-calling baselines, and provides ablations on SFT vs. RL, reward components, tool backbones, and maximum tool-call turns.

Weakness
1. The central empirical claim is weakened by the supervision budget. Although the paper emphasizes that only 8K RefCOCOg samples are used in the RL dataset, the method also relies on a substantial synthetic training pipeline built from SA-1B, including proprietary problem generation with Gemini 2.5 Pro / OpenAI o3, trajectory synthesis with Qwen2.5-VL-72B-Instruct, filtering, and manual curation. This extra task-specific supervision makes the RES comparison difficult to interpret as a clean apples-to-apples evaluation.

2. The analysis of multi-turn behavior is not fully convincing. The paper shows that performance improves as the training-time max-turn budget increases up to 8, and that successful trajectories average around 6 turns, but this does not establish that the learned policy has acquired an efficient adaptive stopping rule. The results may still partly reflect larger interaction budgets rather than a cleaner demonstration of policy quality.

3. The method is technically sound as an engineering pipeline, but the paper does not cleanly isolate how much gain comes from multi-turn interaction itself versus stronger synthetic supervision, larger training recipe complexity, and additional test-time compute.

4. The writing quality is poor. There are multiple grammar and wording issues throughout the paper, and some claims are overstated relative to the actual technical novelty.

5. The paper repeatedly frames the method as “agentic RL”, but the concrete action space is fairly constrained. In my view, the system is more accurately described as a multi-turn tool-calling segmentation policy than a genuinely new agentic RL framework.

---

> ### Author Rebuttal · Authors · 2026-03-30
>
> Thank you for the detailed and careful review. We appreciate that you found the problem setting technically reasonable and the empirical section fairly extensive. Below we respond point-by-point to your key questions, and then address the additional concerns raised in the weakness section.
>
> **Q1: This paper is built from familiar ingredients: synthetic problem generation, trajectory synthesis and filtering, cold-start SFT, and GRPO-style RL with shaped rewards...**
>
> Prior text-guided segmentation methods usually formulate the task as a **single forward pass**, where the model predicts pixel prompts or masks in one shot, which does not fully exploit the perceptual capability of VLMs.
>
> Inspired by ReAct-style agent paradigms, we design **RSAgent** as a **multi-turn tool-using framework**, which addresses the limitations of prior one-shot methods by enabling iterative verification, re-localization, and refinement through external visual tools and returned visual feedback.
>
> More precisely, our contribution is to formulate **text-guided segmentation as a multi-turn reason–act–observe decision process with external visual tools**, and to show that dense-prediction-specific post-training can optimize such a policy for accurate **pixel-level** decision making. In this sense, RSAgent differs from prior ReAct-style work by focusing on **fine-grained pixel-level grounding and segmentation**, rather than textual answer generation.
>
> To make this clearer, in the revision we will replace stronger wording such as **"agentic RL framework"** with more precise descriptions such as **"multi-turn tool-using segmentation policy**.
>
> ---
>
> **Q2: How much of the reported gain comes from the proposed multi-turn policy itself...**
>
> We agree this is the most important controlled comparison. To address it directly, we add a stronger baseline by training a **single-pass model with the same synthetic cold-start pipeline**, and further extend it with the same RL recipe (test-time scaling), so that **single-pass post-training** and **multi-turn post-training** can be compared under much more comparable supervision.
>
> **Results (controlled comparison).**
>
> | Method | ReasonSeg test gIoU | ReasonSeg test cIoU | RES testA gIoU | RES testA cIoU |
> |---|---:|---:|---:|---:|
> | Base model (single-pass) | 50.1 | 43.9 | 64.4 | 64.9 |
> | Single-pass + same synthetic SFT pipeline | 57.5 | 52.2 | 72.7 | 73.5 |
> | Single-pass + same synthetic RL pipeline | 61.3 | 53.1 | 69.6 | 70.1 |
> | RSAgent (multi-turn, SFT only) | 61.2 | 55.4 | 73.3 | 73.9 |
> | RSAgent (multi-turn, RL only) | 62.0 | 54.3 | 76.1 | 77.3 |
> | RSAgent (multi-turn, SFT + RL) | 66.5 | 57.9 | 81.3 | 81.8 |
>
> Under matched synthetic supervision, the multi-turn policy still consistently outperforms its single-pass counterpart. This makes the gain difficult to attribute solely to the additional SA-1B-derived supervision, recipe complexity, or stronger training signals, and supports that multi-turn interaction itself contributes materially to the final performance.
>
> ---
>
> **Q3: The current max-turn analysis does not yet show whether the model stops efficiently when it has high confidence. Does the learned policy truly acquire an adaptive stopping behavior?**
>
> We agree that the original max-turn analysis alone is not sufficient to establish adaptive stopping. To address this concern, we now add:
> - **average turn / efficiency / stopping statistics on ReasonSeg benchmark**,
> - and **turn-wise refinement analysis** showing how mask quality evolves across the interaction trajectory.
>
>
> **Results (efficiency / stopping statistics on ReasonSeg benchmark).**
>
> | Method | Avg. total turns | Avg. segmentation calls | Avg. total inference time (s) | Avg. stop turn |
> |---|---:|---:|---:|---:|
> | RSAgent (max turns = 6) | 4.5 | 3.1 | 28.7 | 3.9 |
> | RSAgent (max turns = 10) | 5.3 | 4.2 | 36.6 | 4.1 |
>
> **Results (turn-wise refinement).**
>
> | Turn | ReasonSeg test gIoU | ReasonSeg test cIoU | RES testA gIoU | RES testA cIoU |
> |---|---:|---:|---:|---:|
> | Turn 1 | 56.1 | 52.6 | 73.7 | 76.0 |
> | Turn 2 | 59.3 | 55.2 | 80.1 | 80.6 |
> | Turn 3 | 62.7 | 57.1 | 80.9 | 81.3 |
> | Turn 4 | 65.3 | 58.7 | 81.0 | **81.6** |
> | Turn 5 | 65.1 | 58.3 | **81.1** | 81.4 |
> | Turn 6 | **66.3** | **57.6** | 79.4 | 80.1 |
> | Turn 7 | 63.4 | 56.4 | 79.2 | 79.8 |
> | Turn 8 | 61.5 | 54.3 | 77.5 | 78.4 |
>
> **Additional trajectory statistics.**
>
> | Statistic | ReasonSeg test | RES testA |
> |---|---:|---:|
> | Avg. best-so-far improvement from Turn 1 to final | 39.1 | 30.7 |
> | % samples improved after Turn 1 | 33.4 | 26.1 |
> | % low-quality first-step predictions rescued by later turns | 11.3 | 8.3 |
>
> These analyses more directly test whether RSAgent performs meaningful iterative refinement rather than simply consuming a larger interaction budget. As the number of max-turns increases, RSAgent doesn't simply make extra turns of tool calls, instead it adaptively stops. Besides, as turn-number increases, the performance worsens.

---

> > ### Author Rebuttal · Reviewer_Qqse · 2026-04-03
> >
> > Thank you for the authors’ rebuttal. I carefully read the additional experiments and analyses. The authors have addressed Q2 and Q3. However, I still feel that the overall pipeline is overly engineering-driven and does not yet demonstrate sufficiently distinctive innovation for this problem. As a suggestion, I noticed that RSAgent (multi-turn, RL-only) achieves a fairly strong success rate. In my view, it has enough potential to serve as an alternative to the cold-start setting, which could naturally avoid many of the fairness concerns related to data. Of course, it may also be the case that current VLMs still lack sufficient tool-use capability, but framing the paper from that perspective would clearly make it more interesting. In consideration of the authors’ effort, I will raise my score.

---

> > > ### Author Response · Authors · 2026-04-06
> > >
> > > We thank Reviewer Qqse for the careful consideration of our rebuttal and for raising the suggestion of fine-tuning the model with RL only. We greatly appreciate your acknowledgment that our clarifications and additional experiments partially resolved your concerns, and we are especially grateful that you raised your score, reflecting improved confidence in our work.
> > >
> > > Our RL-only results do suggest that this is a meaningful perspective. At the same time, prior works (e.g. [1] [2]) have shown tool-integrated RL can suffer from interaction collapse, and that cold-start SFT helps establish a tool-using prior that improves downstream exploration and performance.
> > >
> > > In our current setup, RL-only shows promising performance, but it is still less stable and clearly weaker than cold-start + RL. It can already learn a fairly strong multi-turn policy, but cold-start still makes tool use, action sequencing, and iterative refinement more reliable.
> > >
> > >
> > > **References**
> > > [1] ReTool: Reinforcement Learning for Strategic Tool Use in LLMs
> > > [2] ASTER: Agentic Scaling with Tool-integrated Extended Reasoning

---

### Official Review · Reviewer_MiRH · 2026-03-12

**Soundness:** 3
**Presentation:** 3
**Significance:** 3
**Originality:** 1
**Overall Recommendation:** 4
**Confidence:** 3

**Summary:**

The paper introduces RSAgent, an agentic pipeline powered by a Multimodal Large Language Model (MLLM) designed for text-guided object segmentation.RSAgent addresses two primary challenges: Referring Expression Segmentation(RES) (e.g., "the person with the blue hat") and Reasoning Segmentation (e.g.,"the item crucial for personal safety"), where segmentation queries require complex semantic understanding.
Departing from traditional static approaches, RSAgent utilizes multi-turn agentic reasoning and tool-calling capabilities. The MLLM dynamically interacts with the image by choosing to zoom in on specific regions, rotate the input, or generate point/bounding-box prompts for SAM2. In this multi-turn framework, the model iteratively reviews the output of its tool calls—specifically its previous segmentation attempts—and refines them to produce a precise final mask.
Key Contributions:
- Automated Data Pipeline: A novel "cold-start" procedure to generate supervised fine-tuning (SFT) trajectories using the SA-1B dataset.
- Reinforcement Learning Framework: A specialized RL procedure that enhances segmentation accuracy through a unique triple-objective reward system.
- SOTA Performance: RSAgent achieves state-of-the-art results across both RES and reasoning segmentation tasks on the RefCOCO and ReasonSeg benchmarks, outperforming a wide range of existing methods.

**Compliance With Llm Reviewing Policy:**

Affirmed.

**Final Justification:**

The rebuttal addressed my concerns, so I upgraded my final decision.

**Key Questions For Authors:**

Can you provide a performance breakdown (e.g., mIoU/cIoU) comparing the Base Model, the model after SFT only, and the final model after Reinforcement Learning to isolate the gains of each stage?

To evaluate the impact of the data distillation, how does the performance of the teacher model (Qwen-2.5-VL-72B) on these benchmarks compare to the results achieved by your student model?

Can you provide a table or schematic that grounds the mathematical notation(e.g., $O_t, P_{1:T}, r_t$) with concrete examples from a typical multi-turn reasoning segmentation task?

**Limitations:**

Given the rapid pace of this field, specific contributions may soon be superseded by more powerful MLLMs like Qwen3-VL or advanced segmentation tools such as SAM3. Nevertheless, the underlying methodology remains robust; it serves as a compelling framework for integrating MLLMs with specialized tools through reinforcement learning, ensuring the approach is not strictly dependent on a specific model. However, my main concern is that the novelty remains limited, which is the primary issue for me. Furthermore, the current data pipeline is heavily anchored to the capabilities of Qwen2.5-VL-72B, a reliance that should be further clarified by the authors as requested.

**Strengths And Weaknesses:**

Strengths
- The paper addresses a well-established and highly relevant task within the field.
- The proposed approach is elegant in its simplicity yet yields highly satisfactory results, as evidenced by both quantitative metrics and qualitative visualizations.
- The manuscript is logically structured, clearly written, and remains easy to follow throughout.
- The evaluation is sound and comprehensive, spanning two distinct benchmarks against a wide array of competitive baselines.
- The framework achieves state-of-the-art results, demonstrating its superiority over existing methods.

Weaknesses
- Limited Methodological Innovation: The technical contribution appears somewhat incremental. The combination of Supervised Fine-Tuning (SFT) and Reinforcement Learning (RL) to enhance task-specific MLLM performance is a well-established paradigm. As presented, the framework leans more toward an engineering-heavy solution than a distinct methodological breakthrough.

- Unclear Baseline Performance & Data Dependency: The paper lacks a clear justification for its heavy reliance on SFT data distilled from Qwen-2.5-VL-72B. Crucially, the authors do not report the baseline performance of the underlying model on these segmentation tasks, making it difficult to gauge the true value-add of the proposed pipeline.

-Insufficient Ablation of Training Stages: The specific gains attributed to each phase of the training recipe are not clearly decoupled. To demonstrate the efficacy of the approach, the paper should provide a step-by-step performance breakdown: Initial Baseline vs. Post-SFT vs. Post-RL.

- Notation and Formalism: The clarity of the Methodology section could be significantly improved. The current mathematical notation (e.g., $O_t$,$P_{1:T}$, $r_t$) lacks immediate grounding. Including a dedicated table or a schematic figure—potentially in the supplementary materials—that maps these symbols to concrete examples within the agentic loop would greatly assist the reader.

---

> ### Author Rebuttal · Authors · 2026-03-30
>
> Thank you for the positive assessment of the motivation, writing clarity, empirical breadth, and empirical performance of our method.
>
> **On novelty.**
> Prior text-guided segmentation methods usually formulate the task as a **single forward pass**, where the model predicts pixel prompts or masks in one shot, which does not fully exploit the perceptual capability of VLMs.
>
> Inspired by ReAct-style agent paradigms, we design **RSAgent** as a **multi-turn tool-using framework**, which addresses the limitations of prior one-shot methods by enabling iterative verification, re-localization, and refinement through external visual tools and returned visual feedback.
>
> More precisely, our contribution is to formulate **text-guided segmentation as a multi-turn reason–act–observe decision process with external visual tools**, and to show that dense-prediction-specific post-training can optimize such a policy for accurate **pixel-level** decision making. In this sense, RSAgent differs from prior ReAct-style work by focusing on **fine-grained pixel-level grounding and segmentation**, rather than textual answer generation.
>
> **Question1 on Base / SFT / RL progression.**
> This ablation is already reported in **Table 3**, which displays the performance of SFT-only model, RL-only model, the complete model and training-free models.
>
> **Question2 on the teacher model and distilled supervision.**
> We agree that the role of Qwen-2.5-VL-72B should be explained more clearly. It is used in data construction rather than as a test-time oracle.
>
> **Results (teacher reference under the same framework).**
>
> | Method | ReasonSeg test gIoU | ReasonSeg test cIoU | RES testA gIoU | RES testA cIoU |
> |---|---:|---:|---:|---:|
> | Qwen2.5-VL-72B (same inference framework, zero-shot) | 53.1 | 49.7 | 70.3 | 70.9 |
> | RSAgent (7B) | 66.5 | 57.9 | 81.3 | 81.8 |
>
> This comparison shows that while the large teacher is useful during data construction, the final task-specialized student substantially outperforms the zero-shot teacher under the same interface, indicating that the final gain is not merely inherited from teacher scale at inference time.
>
> Further, to better isolate the gain from multi-turn policy itself, we add a stronger controlled comparison **in our reply to reviewer Qqse's Question2**.
>
>
> **Question3 notation / formalism.**
> Thank you for this helpful suggestion. We provide a compact table below and will include a full notation table plus a schematic multi-turn example in the revision.
>
> | Symbol | Meaning in RSAgent | Concrete example in one episode |
> |---|---|---|
> | $I$ | Input image | An image containing a person holding a cup |
> | $Q$ | Natural-language segmentation query | “Segment the cup held by the person on the left.” |
> | $O_t=(V_t,Q,C_t)$ | Observation at step $t$ | At $t=2$, the agent sees the original image, a zoomed crop from the previous step, the query $Q$, and the accumulated text/tool history so far |
> | $V_t$ | Visual views available at step $t$ | Original image + an overlay showing the previous SAM2 mask + a zoomed-in crop around the candidate region |
> | $C_t$ | Text context accumulated up to step $t$ | System prompt, user query, previous reasoning text, tool calls, and tool feedback |
> | $r_t$ | Intermediate reasoning text generated by the model at step $t$ (**not** the scalar RL reward) | “The cup is small and partially occluded; I should zoom into the hand region before segmenting.” |
> | $a_t$ | Parameterized tool action(s) at step $t$ | zoom_in(bbox=[x1,y1,x2,y2]) or sam2_segment(points=[...], bboxes=[...]) |
> | $o_t$ | Tool output / new visual feedback returned at step $t$ | A zoomed image patch, a rotated image, or a candidate mask produced by SAM2 |
> | $P_{1:T}$ | Full multi-turn trajectory up to termination | A sequence of observations, reasoning text, tool calls, and returned tool outputs across all turns |
>
>
> A typical trajectory looks as follows:
>
> 1. **Step 1.** The agent receives $O_1=(V_1,Q,C_1)$, where $V_1$ contains the original image and $C_1$ contains the initial system/user prompts.
>    The model generates reasoning text $r_1$ such as: “The queried cup is small and attached to the left person; I should first zoom into the left-hand area.”
>    It then emits an action $a_1$={zoom\_in}.
>
> 2. **Step 2.** The toolbox returns $o_1$, i.e., a zoomed crop, which is added into the next observation $O_2$.
>    Based on $O_2$, the model generates $r_2$, then emits $a_2$={sam2\_segment}.
>
> 3. **Step 3.** The toolbox returns $o_2$, i.e., a candidate mask overlay, which becomes part of $O_3$.
>    The model may further refine the mask or terminate if the result is already satisfactory.
>
> 4. **Termination.** The full trajectory $P_{1:T}$ therefore records the multi-turn process of **observe $\rightarrow$ reason $\rightarrow$ act $\rightarrow$ receive feedback**, after which the agent outputs the final mask.

---

> > ### Author Rebuttal · Reviewer_MiRH · 2026-04-07
> >
> > The authors have addressed my concerns.

---

> > > ### Author Response · Authors · 2026-04-07
> > >
> > > Thank you for the careful follow-up and for taking the time to reconsider our rebuttal. We are very glad that our clarifications and additional experiments were able to address your concerns. Since these points have now been resolved, we would be grateful if you could consider upgrading your score accordingly. We remain happy to clarify any further point if helpful.

---

### Decision · Program_Chairs · 2026-04-30

**Decision:**

Accept (regular)

**Comment:**

I recommend Weak Accept. All four reviewers give positive final recommendations, and there is no substantial disagreement in the final assessment. Reviewers generally agree that the paper addresses a relevant problem, that the multi-turn reason–act–observe formulation for text-guided segmentation is well motivated, and that the empirical results are strong, with broad evaluation and competitive performance on both RES and ReasonSeg benchmarks.

The main reservation across reviews is not about soundness, but about novelty and positioning. Several reviewers view the work as more engineering-heavy than algorithmically novel, built from familiar ingredients such as synthetic data generation, SFT, RL post-training, and tool calling. There were also concerns about the role of proprietary data generation, efficiency, and how much of the gain comes from the multi-turn policy itself rather than the broader training recipe.

In the rebuttal, the authors addressed many of these concerns with useful additional evidence, including controlled comparisons against single-pass variants under matched synthetic supervision, analyses of turn-wise refinement and stopping behavior, efficiency statistics, recovery from imperfect intermediate segmentations, and additional clarification on teacher dependence and generalization. While some concerns about conceptual novelty remain, the final reviewer consensus is uniformly positive at the weak-accept level. I also note that the concern raised earlier about a mismatch between reviewer acknowledgements and displayed scores no longer applies: the relevant reviewers’ final displayed overall recommendations are now also 4 (Weak Accept). Overall, I therefore follow the reviewer consensus and recommend acceptance.